# Generative Models: What Do They Know? Do They Know Things? Let's Find Out!

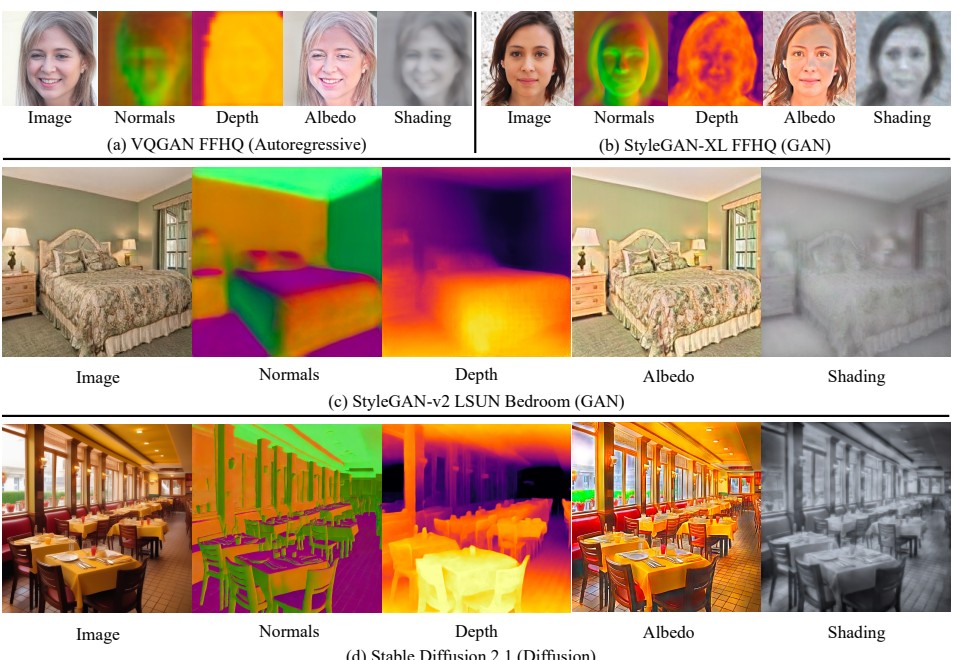

Figure 1: Generative models of various types—Autoregressive, GANs and Diffusion—implicitly encode intrinsic images as a by-product of generative training. We show that a model-agnostic approach, Low-Rank Adaptation (LoRA), can recover this intrinsic knowledge. Applying targeted, lightweight LoRA to attention layers in VQGAN (a) and Stable Diffusion (d), and affine layers in StyleGAN (b and c), allows us to recover fundamental intrinsic images—normals, depth, albedo and shading—directly from the models' learned representations, eliminating the need for additional task-specific decoding heads or layers.

## Abstract

Generative models excel at mimicking real scenes, suggesting they might inherently encode important intrinsic scene properties. In this paper, we aim to explore the following key questions: (1) What intrinsic knowledge do generative models like GANs, Autoregressive models, and Diffusion models encode? (2) Can we establish a general framework to recover intrinsic representations from these models, regardless of their architecture or model type? (3) How small can the required learnable parameters and labeled data be to successfully recover this knowledge? (4) Is there a direct link between the quality of a generative model and the accuracy of the recovered scene intrinsics?

Our findings indicate that a small Low-Rank Adaptators (LoRA) can recover intrinsic images—depth, normals, albedo and shading—across different generators (Autoregressive, GANs and Diffusion) while using the same decoder head that generates the image. As LoRA is lightweight, we introduce very few learnable parameters (as few as 0.04% of Stable Diffusion model weights for a rank of 2), and we find that as few as 250 labeled images are enough to generate intrinsic images with these LoRA modules. Finally, we also show a positive correlation between the generative model's quality and the accuracy of the recovered intrinsics through control experiments.

# 1 INTRODUCTION

Generative models can produce high-quality images that are almost indistinguishable from real-world photographs. They appear to profoundly understand the world, capturing object placement, appearance, and lighting conditions. Yet, it remains an open question how these models encode such detailed knowledge, and whether representations of scene intrinsics—such as depth, normals, albedo and shading—exist within these models and can be explicitly recovered, or if these models manipulate abstract representations of the world to generate these images.

**Why study intrinsic knowledge embedded in generative models?** Understanding how generative models produce realistic outputs allows us to model the physical world better computationally, improving both image generation and interpretation across various applications. As we demonstrate in this paper, most generative models inherently encode intrinsic image representations as a byproduct of training on large-scale image data, and these can be easily recovered. By retrieving this embedded knowledge, we can enhance downstream tasks such as relighting, object compositing, and image editing without the need for large labeled datasets or extensive retraining of the models.

Recent work has begun to study this question. Bhattad et al. (2023a) demonstrated that StyleGAN can encode important scene intrinsics. Similarly, Zhan et al. (2023) showed that diffusion models can understand 3D scenes in terms of geometry and shadows. Chen et al. (2023) found that Stable Diffusion's internal activations encode depth and saliency maps that can be extracted with linear probes. Three independent efforts (Luo et al., 2023b; Tang et al., 2023; Hedlin et al., 2023) discovered correspondences in diffusion models. However, these insights often pertain to specific models, leaving a gap in our understanding of whether such encoding is ubiquitous across generative architectures.

**Why study different models?** While diffusion models (Rombach et al., 2022; Saharia et al., 2022), have gained significant attention, other model types like GigaGAN (Kang et al., 2023), CM3leon (Yu et al., 2023), and Parti (Yu et al., 2022) have shown they can produce similarly high-quality images. By investigating this wide range of models, we can create a general framework that not only applies to current generative models but is also adaptable to future developments and emerging architectures. To the best of our knowledge, this paper is the first to study generative models of all types.

**Why develop a general approach?** A general approach ensures broad applicability to emerging generative models. In this context, we find LoRA (Hu et al., 2022) to be highly effective. LoRA can easily recover scene intrinsics across diverse architectures with small parameter updates and data. This general method lays the groundwork for future research that can build on our findings to explore intrinsic knowledge in new generative models. It is important to note that any approach capable of being applied to all generative models with small or no parameter updates and low data requirements is a reasonable and valid choice. *While we have identified one such method (LoRA) in this work, many others could also recover intrinsic representations across diverse generative models.*

**Why do we need slight modification or small data to recover this knowledge?** Ideally, we recover intrinsic knowledge without any new learning, revealing what the model already "knows." But achieving this purely with no learning is hard and non-trivial. Thus, we limit our approach to light fine-tuning, using little labeled data to avoid introducing new knowledge to the model.

Previous approaches, such as Bhattad et al. (2023a), have found codes in StyleGAN's latent space for each intrinsic image, but such disentangled spaces have not yet been identified in models like diffusion and autoregressive models. Recent depth extraction from diffusion models often involves fine-tuning the entire model (Zhao et al., 2023; Ke et al., 2023) or applying linear probing (Chen et al., 2023). Fine-tuning alters the model significantly, transforming it into a new version and potentially compromising its original image-generating capabilities. This raises the question of whether the depth perception was an innate quality of the model or a product of the fine-tuning process. A drawback of linear probing lies in probing each layer independently. As we show linear probes perform poorly, and our application of LoRA suggests that intrinsic information is distributed throughout the network.

**Why analyze the correlation between recovered intrinsics and improved generative models?** If higher-quality generative models consistently produce better intrinsic images, this suggests an alternative paradigm for improving these models. Instead of blindly scaling up with more data and parameters, we could focus on enhancing the model's ability to capture and recover intrinsic properties. This approach could lead to more efficient improvements in model performance, driven by the quality of the intrinsic knowledge embedded within the model.

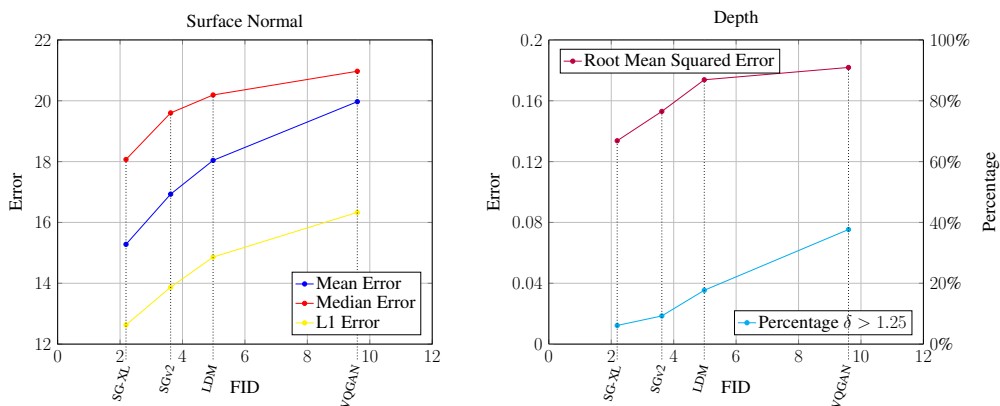

Figure 2: FID vs. metrics of intrinsics recovered from different generative models traind on FFHQ. Enhancements in image generation quality correlate positively with intrinsic recovery capabilities.

Table 1: Summary of scene intrinsics found across different generative models without changing generator head. ✓: Intrinsics can be recovered with high quality. ∼: Intrinsics cannot be recovered with high quality. ×: Intrinsics cannot be recovered.

| Model | Pretrain Type | Domain | Normal | Depth | Albedo | Shading |
|---|---|---|---|---|---|---|
| VQGAN (Esser et al., 2020) | Autoregressive | FFHQ | ∼ | ∼ | ✓ | ✓ |
| SG-v2 (Karras et al., 2020b) | GAN | FFHQ | ✓ | ∼ | ✓ | ✓ |
| SG-v2 (Yu et al., 2021) | GAN | LSUN Bed | ✓ | ✓ | ✓ | ✓ |
| SG-XL (Sauer et al., 2022) | GAN | FFHQ | ✓ | ∼ | ✓ | ✓ |
| SG-XL (Sauer et al., 2022) | GAN | ImageNet | × | × | × | × |
| SD-UNet (single-step) (Rombach et al., 2022) | Diffusion | Open | ✓ | ✓ | ✓ | ✓ |
| $SD_{AUG}$ (multi-step) (Rombach et al., 2022) | Diffusion | Open | ✓ | ✓ | ✓ | ✓ |

We find positive correlations in our experiments between the quality of recovered intrinsics and the improvements in generative model performance. Specifically, we observe this in Stable Diffusion versions 1.1, 1.2 and 1.5, as well as in improved face generators from various GAN and Autoregressive models, as measured by FID. A visual illustration of this correlation is in Fig. 2. These results indicate that higher-quality generators tend to produce more accurate intrinsic representations.

Our contributions are showing that generative models encode intrinsic images across different architectures, including GANs, Autoregressive models and Diffusion models. Our findings are in Tab. 1 and elaborated in Sec. 4. We find a general approach using LoRA to recover these intrinsics, which are competitive, with light fine-tuning and data. This method obtains these properties using the same output head as the original image generation task. Through control experiments, we find a positive correlation between the quality of the generative model and the accuracy of the recovered intrinsics, suggesting that better models naturally produce better intrinsic representations(Fig. 2). This offers a new paradigm for model improvement beyond just scaling data and parameters.

## 2 RELATED WORK

**Generative Models:** Generative Adversarial Networks (GANs) (Goodfellow et al., 2014) have been widely used for generating realistic images. Variants like StyleGAN (Karras et al., 2019), StyleGAN2 (Karras et al., 2020b) and GigaGAN (Kang et al., 2023) have pushed the boundaries in terms of image quality and control. Some work has explored the interpretability of GANs (Bau et al., 2020; Bhattad et al., 2023a), but little is known about their ability to capture scene intrinsics.

Diffusion models (Vincent, 2011; Gutmann & Hyvärinen, 2010) are popular at the moment for generative tasks (Karras et al., 2022; Ho et al., 2020; Rombach et al., 2022). These models have been shown to understand complex scene intrinsics like geometry and shadows (Zhan et al., 2023), but the generalizability of this understanding across different scene intrinsics is largely unexplored.

Autoregressive models (Van Den Oord et al., 2016; Van den Oord et al., 2016) generate images pixel-by-pixel, offering fine-grained control but at the cost of computational efficiency. VQ-VAE-2 (Razavi et al., 2019) and VQGAN (Esser et al., 2020) have combined autoregressive models with vector quantization to achieve high-quality image synthesis. While these models are powerful, their ability to capture and represent scene intrinsics is yet to be investigated.

**Intrinsic Image Recovery:** Barrow & Tenenbaum (1978) highlighted several fundamental scene intrinsics including depth, albedo, shading, and surface normals. A large body of work has focused on extracting some related properties like depth and normals, from images (Eigen et al., 2014; Long et al., 2015; Eftekhar et al., 2021; Kar et al., 2022; Ranftl et al., 2021; Bhat et al., 2023) using labeled data. Labeled albedo and shading are hard to find and as the recent review in Forsyth & Rock (2021) shows, methods involving little or no learning have remained competitive. However, these methods often rely on supervised learning and do not recover intrinsic images from generative models.

Many recent studies have used generative models as pre-trained feature extractors or scene prior learners. They use generated images to enhance downstream discriminative models, fine-tune the original generative model for a new task, learn new layers or decoders to produce desired scene intrinsics (Abdal et al., 2021; Jahanian et al., 2021; Zhang et al., 2021b; Li et al., 2021; Noguchi & Harada, 2020; Bao et al., 2022; Xu et al., 2023; Sariyildiz et al., 2023; Zhao et al., 2023; Ke et al., 2023). InstructCV (Gan et al., 2023) executes computer vision tasks via natural language instructions, abstracting task-specific design choices. However, it requires re-training of the entire diffusion model. In contrast, we show that many generative models capture intrinsic image knowledge implicitly and do not require specialized training to recover this information.

**Knowledge in Generative Models:** Several studies have explored the extent of StyleGAN's knowledge, particularly for 3D information about faces (Pan et al., 2021; Zhang et al., 2021a). Yang et al. (2021) show GANs encode hierarchical semantic information across different layers. Further research has demonstrated that manipulating offsets in StyleGAN can lead to effective relighting of images (Bhattad et al., 2024; 2023b) and extraction of scene intrinsics (Bhattad et al., 2023a). Chen et al. (2023) found internal activations of the LDM encode linear representations of both depth data and a salient-object / background distinction. Wu et al. (2023) also demonstrate rich latent codes of diffusion models can be easily mapped to annotations with small amount of training samples. Tang et al. (2023); Luo et al. (2023b); Hedlin et al. (2023) found correspondence emerges in image diffusion models. Sarkar et al. (2023) showed generative models fail to replicate projective geometry.

Luo et al. (2023a) explored training task-specific "readout" networks to extract signals like pose, depth, and edges from feature maps in Stable Diffusion models for controlling image generation. Our goals are different: We are interested in understanding intrinsic knowledge encoded in these models, while the aim of Luo et al. (2023a) is controlling image generation. Our use of LoRA offers notable advantages in parameter efficiency: itis approximately 5 times more parameter-efficient than readout networks in their application to SD v1-5 (compare 8.5M vs 1.59M). Lastly, the broad applicability of "readout" networks across various generative model types remains uncertain.

A concurrent work Lee et al. (2023) applies a LoRA-like approach to adapt a pre-trained diffusion model for dense semantic tasks. Our work differs from theirs in several aspects: First, their goal is to use pre-trained diffusion models as strong priors for dense prediction. Second, their tasks are within restricted domains, such as bedrooms. Finally, they do not extend to the wide range of generative models our study explores. Our paper not only demonstrates intrinsic knowledge encoded in different architectures but also explores its application in a diverse scene contexts including real images.

## 3 APPROACH

A generative model $G$ maps noise/conditioning information $z$ to an RGB image $G(z) \in \mathbb{R}^{H \times W \times 3}$. We add to $G$ with a small set of parameters $\theta$ that allow us to produce, using the same architecture as $G$, an image-like map with up to three channels, representing scene intrinsics like surface normals.

**Our Framework.** We recover intrinsic properties of an image (such as depth) using a small number of labeled examples (image/depth map pairs) as supervision. In cases where we do not have access to the actual intrinsic properties, we use models trained on large datasets to generate estimated intrinsics (such as estimated depth for an image) as pseudo-ground truth, used as training targets for $G_\theta$. To optimize $\theta$ of $G_\theta$ using a pseudo-ground truth predictor $\Phi$, we minimize the objective:

$$\min_\theta \mathbb{E}_z[d(G_\theta(z), \Phi(G(z)))], \tag{1}$$

where $d$ is a distance metric that depends on the intrinsics we wish to learn.

Diffusion models require special treatment since their input and output are with the same dimension. During inference, diffusion models repeatedly receive a noisy image as input. Thus instead of

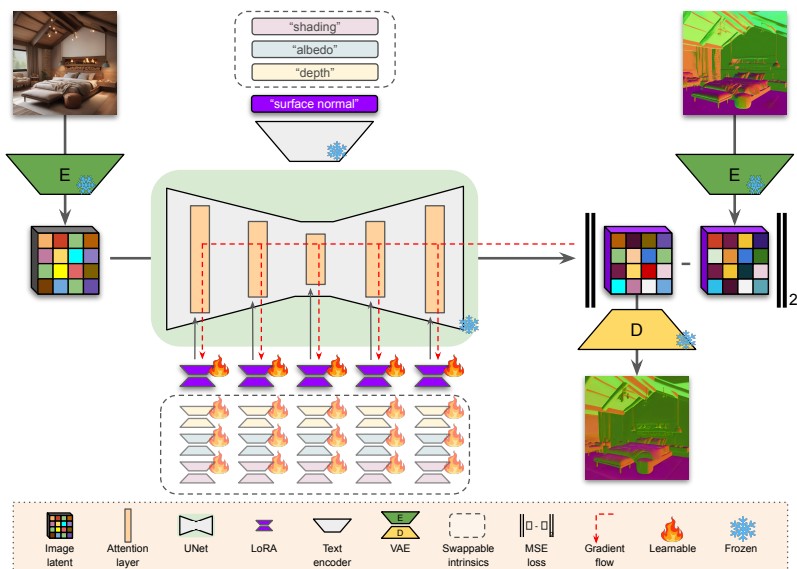

Figure 3: Overview of our framework applied to Stable Diffusion's UNet in a single-step manner. We adopt an efficient fine-tuning approach, low-rank adaptors (LoRA) corresponding to key feature maps – attention matrices – to reveal scene intrinsics. Distinct adaptors are optimized for each intrinsic (*violet* adaptors for surface normals; swappable with other intrinsics). We use a few labeled examples for this fine-tuning and directly obtain scene intrinsics using the same decoder that generates images, circumventing the need for specialized decoders or comprehensive model re-training.

conditioning noise $z$ we feed an image $x$(generated or real) to a diffusion model $G$. In this case, given a real image $x$, our objective function becomes $\min_\theta \mathbb{E}_x[d(G_\theta(x), \Phi(x))]$.

For surface normals $\Phi$ is Omnidatav2 (Kar et al., 2022). To generate pseudo ground truth for depth we use ZoeDepth (Bhat et al., 2023) as the predictor $\Phi$. For Albedo and Shading $\Phi$ is Paradigms (Forsyth & Rock, 2021; Bhattad & Forsyth, 2022). For SG2, SGXL and VQGAN, $d$ in Eq.1 is

$$d(x, y) = 1 - cos(x, y) + \|x - y\|_1 \qquad (2)$$

for normal and MSE for other intrinsics. For latent diffusion, there isn't a clear physical meaning to the relative angle of latent vectors in encoded normals, so we use the standard MSE for all intrinsics.

We use LoRA, a parameter-efficient adaptation technique, to recover image intrinsics from generative models. LoRA introduces a low-rank weight matrix $W^*$, which has a lower rank than the original weight matrix $W \in \mathbb{R}^{d_1 \times d_2}$. This is achieved by factorizing $W^*$ into two smaller matrices $W_u^* \in \mathbb{R}^{d_1 \times d^*}$ and $W_l^* \in \mathbb{R}^{d^* \times d_2}$, where $d^*$ is chosen such that $d^* \ll \min(d_1, d_2)$. The output $o$ for an input activation $a$ is then given by:

$$o = Wa + W^*a = Wa + W_u^*W_l^*a. \qquad (3)$$

To preserve the original model's behavior at initialization, $W_u^*$ is initialized to zero.

**Applying LoRA for diffusion models**, LoRA adaptors are learned atop cross-attention and self-attention layers, which aggregate geometrical information and "reflect the overall composition" of the input (Hertz et al., 2023). The UNet is utilized as a dense predictor, transforming an RGB input into intrinsics in one step. Depending on the intrinsic, the textual input varies among "surface normal", "depth", "albedo", or "shading". Fig. 3 shows our pipeline. For **GANs**, LoRA modules are integrated with the affine layers that map from w-space to s-space (Wu et al., 2021). In the case of **VQGAN, an autoregressive model**, LoRA is applied to the convolutional attention layers within the decoder.

## 4 EXPERIMENTS

In this section, we outline our findings. Sec. 4.1 and Sec. 4.2 demonstrate LoRA's general applicability across generative models and efficiency in terms of parameters and labeling, respectively. In Sec. 4.3, we conduct control experiments and discover a strong correlation between the quality of a generator

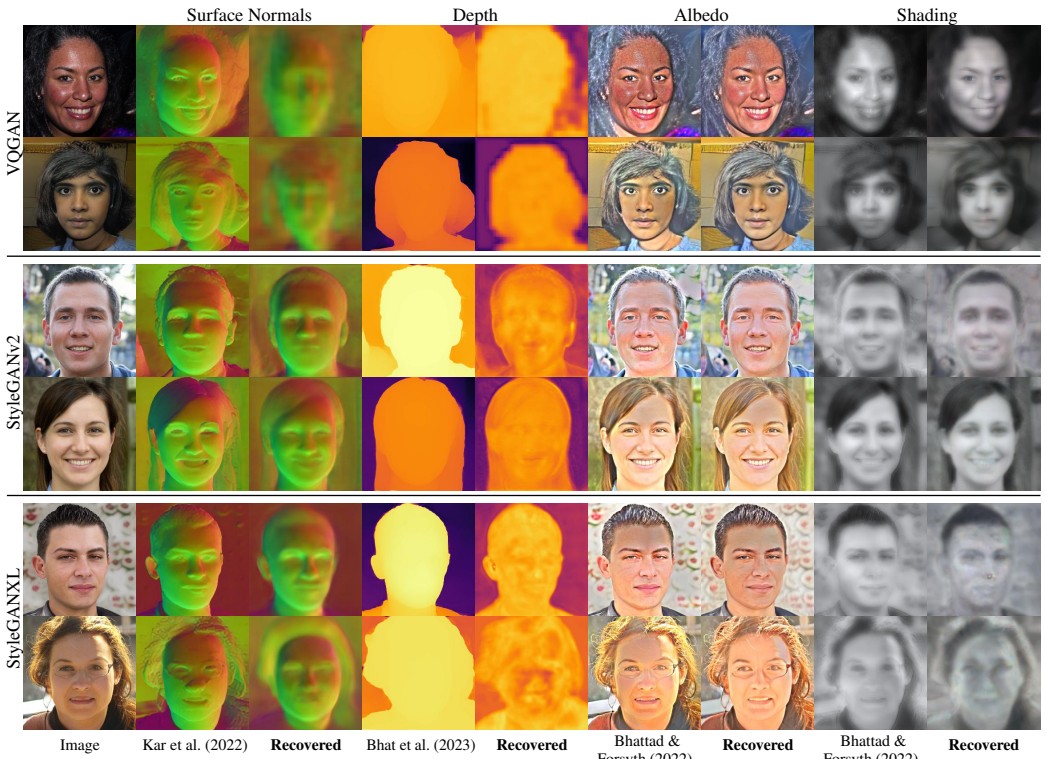

Figure 4: Scene intrinsics from VQGAN, StyleGAN-v2, and StyleGAN-XL – trained on FFHQ dataset: The "image" column shows the synthetic images produced by each model. Other columns show four scene intrinsics predicted by a SOTA non-generative model and recovered by LoRA.

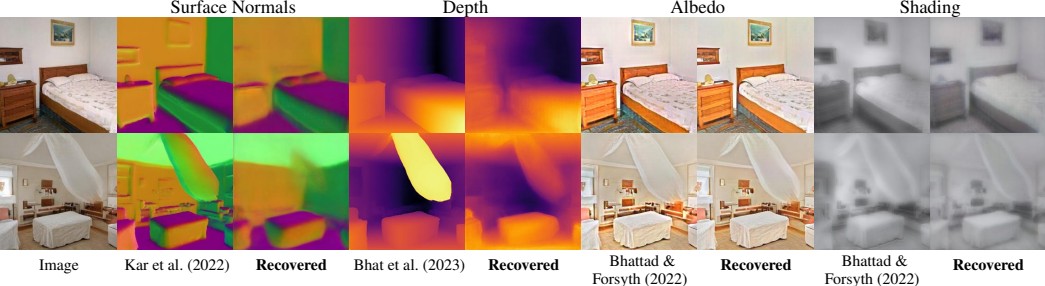

Figure 5: Our recovered scene intrinsics from StyleGAN-v2 trained on LSUN bedroom images.

and the accuracy of its recovered intrinsics (Sec. 4.3). Additional ablation studies and baseline comparisons further confirm LoRA's robustness (Appendix B). Note: our analysis in Sec. 4.2 uses a single-step approach for intrinsic image recovery from stable diffusion. In Sec. 5, we discuss the challenge of naively applying LoRA to a multi-step Stable Diffusion model. To address this, we propose a simple modification to the architecture . We refer to this modified model as $SD_{AUG}$.

### 4.1 FINDING 1: INTRINSIC IMAGES ARE ENCODED ACROSS GENERATIVE MODELS, AND LoRA IS A GENERAL APPROACH FOR RECOVERING THEM

We aim to recover intrinsic images across diverse generative models, including StyleGAN-v2 (Yu & Smith, 2019), StyleGAN-XL (Sauer et al., 2022), and VQGAN (Esser et al., 2020), trained on datasets like FFHQ (Karras et al., 2020b), LSUN Bedrooms (Yu et al., 2015), and ImageNet (Deng et al., 2009). LoRA adapters are tailored to each model and dataset to recover intrinsics: surface normals, depth, albedo, and shading, demonstrating broad applicability and robustness in both qualitative assessments (Fig. 1, 4, 5, 7) and quantitative (Tab. 2 on generated images, Tab. 3 on real images). In all experiments – covering both generated and real images – we use pseudo-ground truth from pre-trained models as a supervisory signal for fine-tuning LoRA adapters to discover scene intrinsics

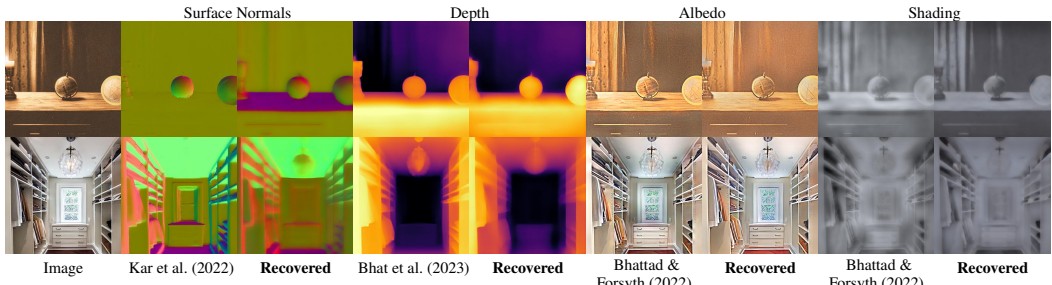

Figure 6: StyleGAN-XL on ImageNet. Recovered surface normals and depth maps, while capturing the basic shape and volume, lack precise detail and display artifacts. Albedo and Shading recoveries fail. These results are correlated with the overall bad image generation quality.

Figure 7: Scene intrinsics recovered from randomly generated stable diffusion images using LoRA. Recovered intrinsics appear to be better. For example, the table's normal in the first row is more accurate compared to Kar et al. (2022). The rightmost globe also appears to be closer to the camera in recovered depth compared to Bhat et al. (2023). In the second row, ceiling lamp normals are visible in recovered intrinsics but not in Kar et al. (2022). These comparison highlights that the recovered intrinsics can closely align with, and sometimes surpass, these supervised monocular predictors.

Table 2: Quantiative analysis of scene intrinsics recovery performance by LoRA on generated images. We compare with pseudo ground truths from Omnidata-v2 for surface normals, ZoeDepth for depth, and Paradigms for albedo and shading. Metrics include mean angular error, median angular error, and L1 error for surface normals; RMS and $\delta < 1.25$ for depth; RMS for albedo and shading.

| Model | Pre-training Type | Domain | LoRA Param. | Surface Normal | | | Depth | | Albedo | Shading |
|---|---|---|---|---|---|---|---|---|---|---|
| | | | | Mean Error°↓ | Median Error°↓ | L1 Error$_{\times 100}$↓ | RMS↓ | $\delta < 1.25_{\times 100\%}$↑ | RMS↓ | RMS↓ |
| VQGAN | Autoregressive | FFHQ | 0.18% | 19.97 | 20.97 | 16.33 | 0.1819 | 62.33 | 0.0345 | 0.0106 |
| StyleGAN-v2 | GAN | FFHQ | 0.57% | 16.93 | 19.60 | 13.87 | 0.1530 | 90.74 | 0.0283 | 0.0110 |
| StyleGAN-XL | GAN | FFHQ | 0.29% | 15.28 | 18.07 | 12.63 | 0.1337 | 93.87 | 0.0287 | 0.0125 |
| StyleGAN-v2 | GAN | LSUN Bedroom | 0.57% | 13.94 | 24.76 | 11.49 | 0.0897 | 66.88 | 0.0270 | 0.0074 |
| StyleGAN-XL | GAN | ImageNet | 0.29% | 24.09 | 25.52 | 19.44 | 0.2175 | 38.38 | 0.1065 | 0.0119 |
| SD$_{AUG}$ (multi step) | Diffusion | Open | 0.17% | 21.41 | 28.57 | 17.39 | 0.2042 | 41.21 | 0.0881 | 0.0099 |
| SD-UNet (single step) | Diffusion | Open | 0.17% | 16.63 | 23.64 | 13.69 | 0.1179 | 52.59 | 0.0487 | 0.0118 |

within generative models as previously mentioned in Sec. 3. We use LoRA with Rank 8 as default for all generative models if not otherwise mentioned.

We find LoRA can recover intrinsic images from almost all models tested. The notable exception is StyleGAN-XL trained on ImageNet, where it yields qualitatively poor results, which we attribute to the model's limited ability to generate realistic images (Fig. 6). This suggests the recovered intrinsic quality is correlated with the generative model's fidelity (see Sec. 4.3). For evaluating generated images, we benchmarked against pseudo-ground truths derived from existing models, compensating for the lack of true ground truths. The performance, gauged through these comparisons, provides useful indicators but must be interpreted within the context of the selected pseudo-ground truths.

Thanks to their architecture as image-to-image translators, diffusion models are powerful image generators that easily apply to real images. Exploiting this, we use LoRA to directly retrieve intrinsic images from Stable Diffusion's UNet in a single step, bypassing the iterative reverse denoising process. The model takes a real image as input and outputs its intrinsic components, allowing for direct evaluation against actual ground truth. on DIODE dataset (Vasiljevic et al., 2019). We use the official training/evaluation split in all of our DIODE experiments. For training with fewer samples, we randomly chose samples from the official training partition. All the metrics we reported on DIODE are computed over the entire evaluation set. In Tab. 3, we find that the LoRA adapters not only

Table 3: Quantitative analysis of recovered scene intrinsics across different models on real images.

| Model | Pre-training Type | LoRA Param | Surface Normal | | | Depth | |
|---|---|---|---|---|---|---|---|
| | | | Mean Error°↓ | Median Error°↓ | L1 Error$_{\times 100}$↓ | RMS↓ | $\delta < 1.25_{\times 100}$↑ |
| Omnidata-v2 (Kar et al., 2022)/ZoeDepth (Bhat et al., 2023) | Supervised | - | **18.90** | 13.36 | **15.21** | 0.2693 | **47.56** |
| DINOv2 | Non-Generative | 0.26% | 19.74 | 13.72 | 16.00 | 0.2094 | 44.32 |
| SD$_{AUG}$ (multi step) | Diffusion | 0.17% | 23.74 | 19.08 | 19.31 | 0.2651 | 43.19 |
| SD-UNet (single step) | Diffusion | 0.17% | 20.31 | **12.54** | 16.53 | **0.2046** | 44.90 |

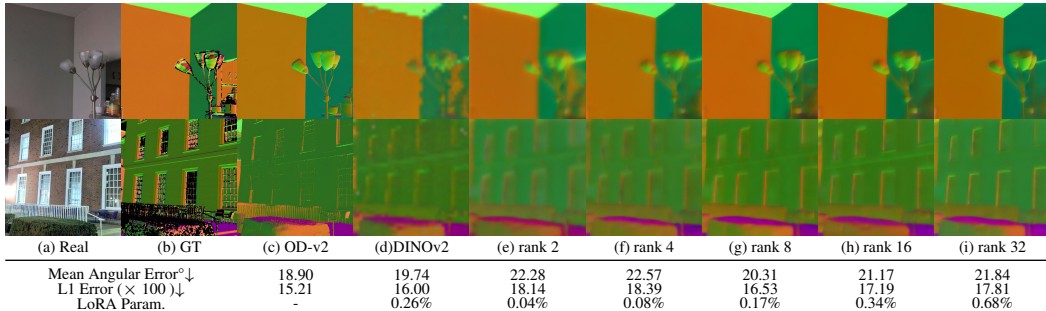

| | (a) Real | (b) GT | (c) OD-v2 | (d)DINOv2 | (e) rank 2 | (f) rank 4 | (g) rank 8 | (h) rank 16 | (i) rank 32 |
|---|---|---|---|---|---|---|---|---|---|
| Mean Angular Error°↓ | | 18.90 | 19.74 | 22.28 | 22.57 | 20.31 | 21.17 | 21.84 |
| L1 Error (× 100)↓ | | 15.21 | 16.00 | 18.14 | 18.39 | 16.53 | 17.19 | 17.81 |
| LoRA Param. | | - | 0.26% | 0.04% | 0.08% | 0.17% | 0.34% | 0.68% |

Figure 8: Parameter Efficiency of LoRA. We evaluate various rank settings for normals recovery. Lower ranks such as 8 offer a balance between efficiency and effectiveness. All model variants are trained using SD's UNet (v1.5) with 4000 samples. Performance metrics, such as Mean Angular Error and L1 Error for normals, and additional parameter counts are detailed below each variant.

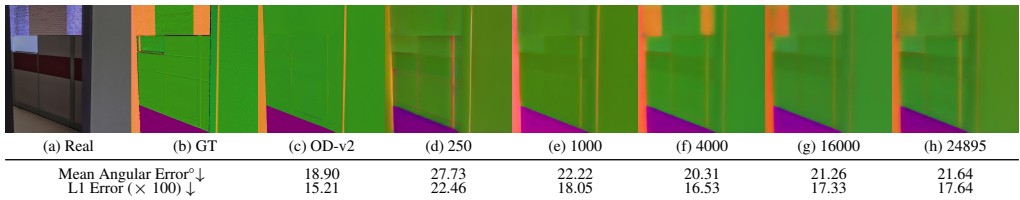

| | (a) Real | (b) GT | (c) OD-v2 | (d) 250 | (e) 1000 | (f) 4000 | (g) 16000 | (h) 24895 |
|---|---|---|---|---|---|---|---|---|
| Mean Angular Error°↓ | | 18.90 | 27.73 | 22.22 | 20.31 | 21.26 | 21.64 |
| L1 Error (× 100)↓ | | 15.21 | 22.46 | 18.05 | 16.53 | 17.33 | 17.64 |

Figure 9: Data efficiency. Note: SOTA supervised model (c), was trained using 12M+ labeled training samples. Even with 250 samples, LoRA captures surface normals. We observe the best performance with 4k samples. Models (d)-(h) all use the same SD UNet(v1-5) and rank 8 LoRA.

matches but, in several metrics (median error for surface normals, RMSE for depth), surpasses the performance of Omnidata and ZoeDepth – the source of its training signal – while using significantly less data, parameters, and training time (see Sec.4.2).

**Extending to DINO.** LoRA intrinsic recovery extends beyond generative models to self-supervised, non-generative models like DINO (Darcet et al., 2023). We apply linear head and LoRA modules following Oquab et al. (2023) to project DINO features into pixel space. Using DINOv2's 'giant' model, we find quantitative results comparable to those from Stable Diffusion, with only a 0.26% increase in parameters. But DINOv2 tends to recover intrinsics with visible discontinuities (Fig. 8d).

### 4.2 FINDING 2: TINY NEW PARAMETERS & DATA ARE ENOUGH FOR INTRINSIC RECOVERY

Our single-step SD-UNet model, distinguished by its high quantitative performance, serves as the basis for ablations that assess the influence of rank and labeled data quantity on intrinsic recovery efficiency. We verify that our requirements for compute, parameters, and data are low.

**Parameter efficiency.** Fig. 8 shows normal predictions across LoRA ranks. The best accuracy is achieved with Rank 8, which also generalizes to other intrinsics and models. Notably, a Rank 2 LoRA with only 0.4M additional parameters (a mere 0.04% increase) still yields good performance. Note that across different models, Rank 8 adds only 0.17% to 0.57% additional parameters (Tab. 2).

**Label efficiency.** Ablations of labeled data size is included in Fig. 9. Peak performance is reached by using a modest 4000 training examples, with credible predictions visible from as few as 250 samples.

### 4.3 FINDING 3: BETTER THE GENERATOR BETTER IS INTRINSIC IMAGE RECOVERY

To assess if our method leverages pre-trained generative capabilities or primarily depends on LoRA layers, we performed a control experiment using a randomly initialized SD UNet, following the same

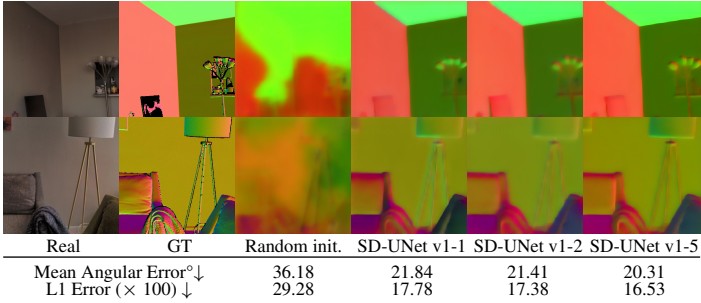

| | Real | GT | Random init. | SD-UNet v1-1 | SD-UNet v1-2 | SD-UNet v1-5 |
|---|---|---|---|---|---|---|
| Mean Angular Error°↓ | | | 36.18 | 21.84 | 21.41 | 20.31 |
| L1 Error (× 100)↓ | | | 29.28 | 17.78 | 17.38 | 16.53 |

Figure 10: Better generators encode better intrinsics. We compare different versions of Stable Diffusion (v1-1, v1-2, v1-5). The progress from SD v1-1 to SD v1-5 shows improvements in recovered intrinsics paralleling improvements in image generation. Control experiments with a randomly initialized UNet fail to retrieve surface normals, emphasizing the reliance on learned priors from generative training for effective intrinsic representation recovery.

training protocol of our SD-UNet model. The poor results from this model (see Fig. 10) corroborate that the learned features developed during generative pre-training are crucial for intrinsic retrieval, rather than the LoRA layers alone. Furthermore, analyzing different Stable Diffusion versions (v1-1, v1-2 and v1-5) under the same training protocol reveals that enhancements in image generation quality correlate positively with intrinsic recovery capabilities. This assertion is further reinforced by observing a correlation between lower FID scores (9.6 for VQGAN (Esser et al., 2020), 3.62 for StyleGAN-v2 (Karras et al., 2020a) and 2.19 for StyleGAN-XL (Sauer et al., 2022)) and improved intrinsic predictions in our FFHQ experiments (Fig. 4 and Tab. 2: first three rows), confirming that superior generative models yield more accurate intrinsics.

## 4.4 FINDING 4: LoRA RECOVERS BETTER INTRINSIC IMAGES THAN OTHER APPROACHES

We compare LoRA with two common approaches: linear probing and full model fine-tuning. Following Chen et al. (2023) for linear probing and using standard fine-tuning practices, we train all methods with a small dataset of 250 samples to 16000 samples. All three are trained with the same number of epochs and have converged at the end of the training. Our findings in Tab. 4 and Fig. 11 show that LoRA significantly outperforms the other two in low-data regimes, validating its preferable efficacy and data efficiency.

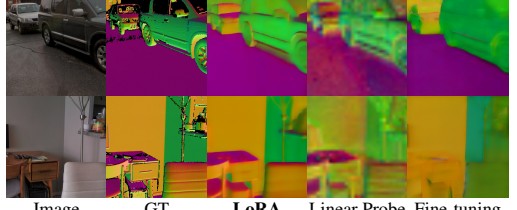

| Image | GT | **LoRA** | Linear Probe | Fine-tuning |

Figure 11: LoRA recovers better intrinsics. Here all approaches use 250 labeled data.

Table 4: We find LoRA to consistently outperform baselines for different training samples.

| | Steps/s | Peak Train GPU Mem% | 250 | | 1000 | | 4000 | | 16000 | |
|---|---|---|---|---|---|---|---|---|---|---|
| | | | Mean Error°↓ | L1 ×100↓ | Mean Error°↓ | L1 ×100↓ | Mean Error°↓ | L1 ×100↓ | Mean Error°↓ | L1 ×100↓ |
| Linear Probe | 2.13 | 29.46% | 29.10 | 23.74 | 28.45 | 23.25 | 28.52 | 23.26 | 28.22 | 23.11 |
| Fine-tuning | 0.77 | 86.78% | 34.40 | 27.58 | 25.19 | 20.28 | 28.03 | 22.17 | 27.39 | 22.24 |
| LoRA | 0.94 | 63.48% | **27.73** | **22.46** | **22.22** | **18.05** | **20.31** | **16.53** | **21.26** | **17.33** |

## 5 TOWARDS IMPROVED INTRINSIC IMAGES RECOVERY

In the previous section, we showed that SD-UNet captures various intrinsic images like normals, depth, albedo, and shading, as evidenced by our evaluation. A natural question arises: can we improve these intrinsics using multi-step diffusion inference? While multi-step diffusion improves sharpness, we find two challenges: (a) intrinsics misaligned with input, and (b) shift in the distribution of outputs relative to the ground truth (visually manifesting as a color shift) (see Fig. 12).

To address (a), we augment the noise input to the UNet with the input image's latent encoding, as in InstructPix2Pix (Brooks et al., 2023). These new parameters are frozen. (b) is a known artifact attributed to SD's difficulty generating images that are not with medium brightness (Deck & Bischoff, 2023; Lin et al., 2023). Lin et al. (2023) propose a Zero SNR strategy that improves color consistency but requires SD trained with a v-prediction objective, absent in SDv1-5. However, SD v2-1 employs a

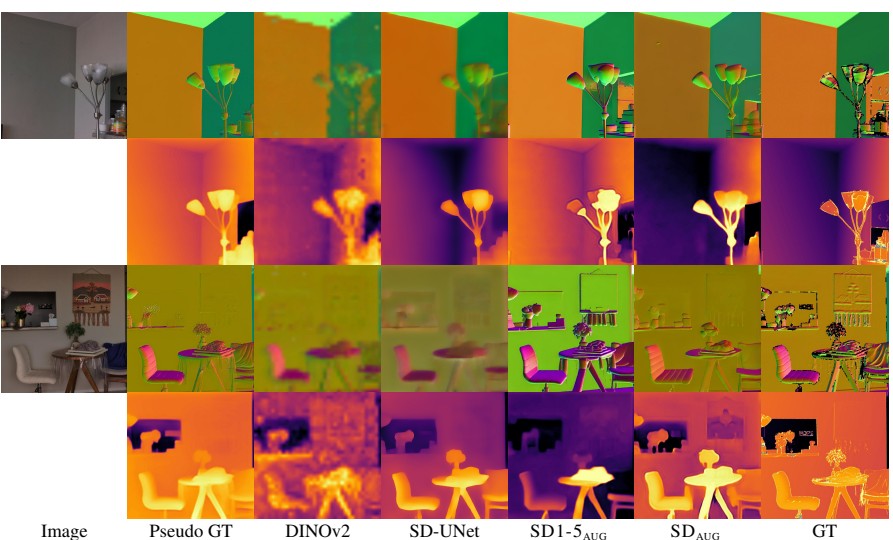

Figure 12: Naive multi-step diffusion leads to wrong intrinsics (fourth column). Our augmentation ($SD_{AUG}$), the fifth column, recovers with the correct layout. The last column further demonstrates highly detailed intrinsic recovery by training LoRA exclusively on domain-specific bedroom images.

Figure 13: We show normals (top in each set) and depth (bottom in each set) derived from improved multi-step diffusion process from $SD_{AUG}$. $SD1\text{-}5_{AUG}$ is similar to $SD_{AUG}$ except it uses SDv1-5 and does not use Zero SNR strategy. $SD1\text{-}5_{AUG}$ presents sharper details, especially in complex areas (lamp stand and chair). $SD_{AUG}$, on the other hand, have a significant improvement in reducing color shifts while maintaining sharpness, as seen in the comparison with ground truth in the last column.

v-prediction objective. Therefore we replace SDv1-5 with SDv2-1 while maintaining our previously described learning protocol. We name this multi-step augmented SDv2-1 model $SD_{AUG}$. $SD_{AUG}$ solves the misalignment issue and reduces the color shift significantly (Fig. 13), resulting in the generation of high-quality, sharp scene intrinsics with improved quantitative accuracy. However, quantitatively, the results still fall short of our single-step SD-UNet result.

## 6 DISCUSSIONS AND LIMITATIONS

We find consistent evidence that generative models implicitly learn intrinsic images, allowing tiny LoRA adapters to recover them with light fine-tuning on small labeled data. More powerful generative models produce more accurate intrinsic images, strengthening our hypothesis that learning this information is a natural byproduct of learning to generate images well.

**Limitations**. Although we show that generative models carry a wealth of intrinsic information, it is still ambiguous how these models use this information when generating images. Secondly, even though our framework is both parameter and label-efficient, we believe there is still room for further reduction and perhaps the development of a parameter-free approach. Lastly, the $SD_{AUG}$ generates sharper results but still lags behind its single-step counterpart in terms of quantitative analysis. Further work is needed to explore this question.

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

# A ADDITIONAL ABLATION STUDIES

## A.1 NUMBER OF DIFFUSION STEPS

| Mean Angular Error°↓ | 25.83 | 23.79 | 23.48 | 23.86 | 23.79 | 23.74 | 23.67 |
|---|---|---|---|---|---|---|---|
| L1 Error (× 100) ↓ | 21.08 | 19.39 | 19.10 | 19.40 | 19.35 | 19.31 | 19.25 |

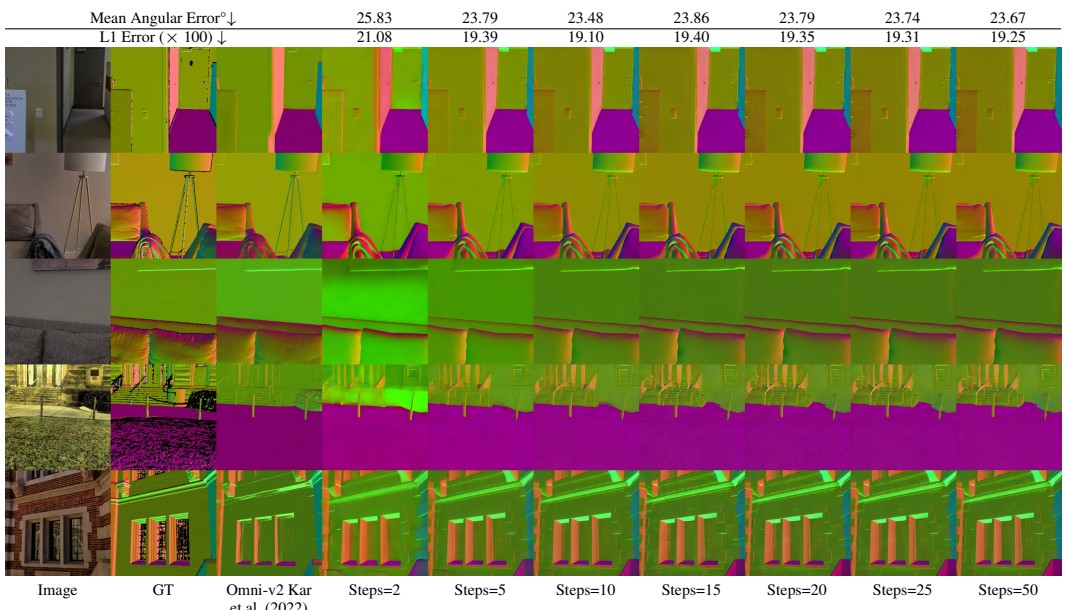

| Image | GT | Omni-v2 Kar et al. (2022) | Steps=2 | Steps=5 | Steps=10 | Steps=15 | Steps=20 | Steps=25 | Steps=50 |

Figure 14: Ablation study to determine the effect of varying numbers of diffusion steps while keeping CFG fixed at 3.0. Our findings show that there are very small differences, both in terms of quantity and quality, after 10 steps. For our main paper, we report results for 25 steps as it is more stable across different intrinsics.

To assess the impact of the number of diffusion steps on the performance of the multi-step $SD_{AUG}$ model, we conducted an ablation study. The results are presented in Fig. 14. For all our experiments in the main text, we used DPMSolver++ (Lu et al., 2022). Interestingly, the quality of results did not vary significantly with an increased number of steps, indicating that 10 steps are sufficient for extracting better surface normals from the Stable Diffusion. Nevertheless, we use 25 steps for all our experiments because it is more stable across different image intrinsics.

## A.2 CFG SCALES

When working with the multi-step $SD_{AUG}$ , the quality of the final output is influenced by the choice of classifier-free guidance (CFG) scales during the inference process. In Fig. 15, we present a comparison of the effects of using different CFG scales. Based on our experiments, we found that using CFG=3.0 results in the best overall quality and minimizes color-shift artifacts.

# B OTHER ABLATIONS AND BASELINES

We extensively study the effect of applying LoRA to different attention layers within Stable Diffusion models. Specifically, we investigate the outcomes of targeting up-blocks, mid-block, down-blocks, cross-attention, and self-attention layers individually. We find (Fig. 16) that isolating LoRA to up or down blocks or the mid-block alone is less effective or diverges, and applying to either cross- or self-attention layers yields decent results, though combining them is best.

Additionally, we evaluated other image editing methods such as Textual Inversion (Gal et al., 2022) and VISII (Nguyen et al., 2023), alongside InstructPix2Pix's response to "Turn it into a surface normal map" instruction (Brooks et al., 2023). As shown in Fig. 17, these methods perform poorly for intrinsic image extraction, demonstrating the effectiveness of the LoRA approach in extracting scene intrinsics.

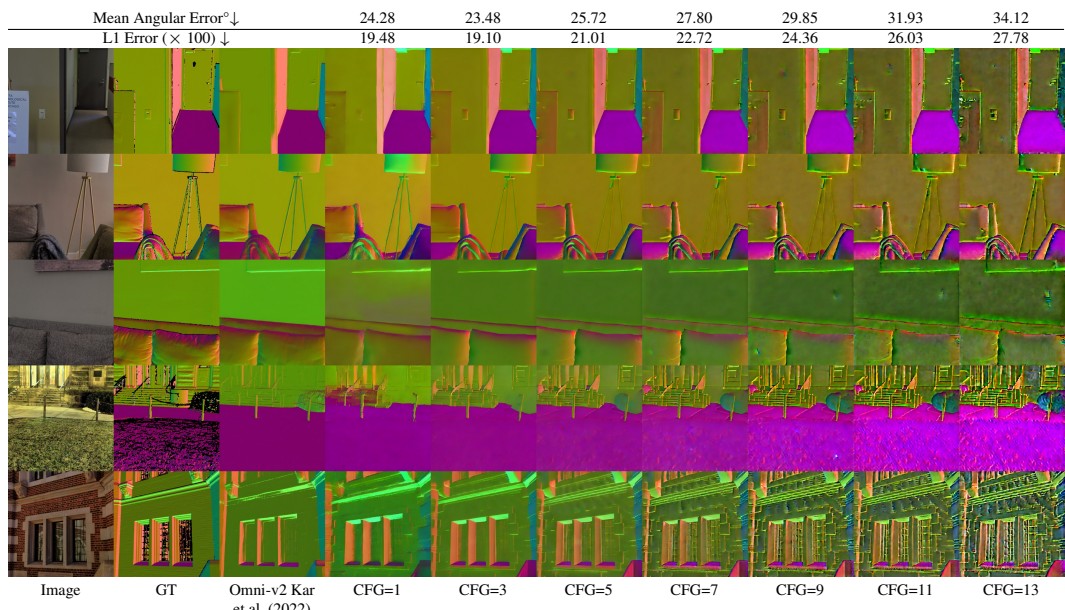

| Mean Angular Error°↓ | | 24.28 | 23.48 | 25.72 | 27.80 | 29.85 | 31.93 | 34.12 |
| L1 Error (× 100) ↓ | | 19.48 | 19.10 | 21.01 | 22.72 | 24.36 | 26.03 | 27.78 |
| Image | GT | Omni-v2 Kar et al. (2022) | CFG=1 | CFG=3 | CFG=5 | CFG=7 | CFG=9 | CFG=11 | CFG=13 |

Figure 15: Ablation study analyzing the impact of different classifier-free guidance (CFG) on SD$_{AUG}$ surface normal prediction. For efficiency, we experimented with a step of 10. We observed that CFG=1 sometimes led to incorrect semantic predictions, particularly in the case of stairs in row 4. On the other hand, using large CFGs (5 and beyond) results in more severe color shift problems.

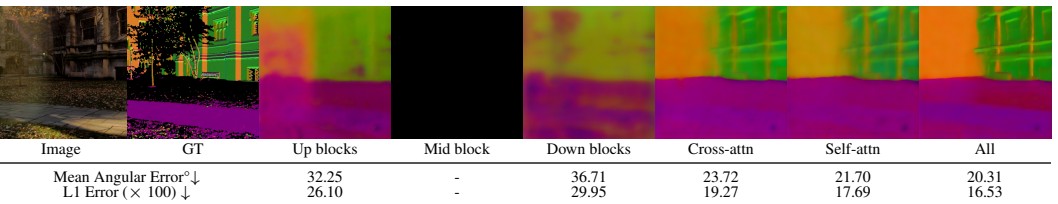

| Image | GT | Up blocks | Mid block | Down blocks | Cross-attn | Self-attn | All |
|---|---|---|---|---|---|---|---|
| Mean Angular Error°↓ | | 32.25 | - | 36.71 | 23.72 | 21.70 | 20.31 |
| L1 Error (× 100) ↓ | | 26.10 | - | 29.95 | 19.27 | 17.69 | 16.53 |

Figure 16: Ablation study on the effect of applying LoRA on different types of attention layers. We started all models with SD v1-5, 4000 training samples and LoRA rank=8. Training with LoRA only on the mid block never converges.

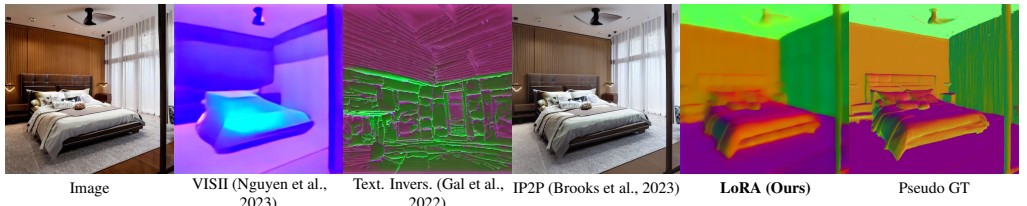

| Image | VISII (Nguyen et al., 2023) | Text. Invers. (Gal et al., 2022) | IP2P (Brooks et al., 2023) | **LoRA (Ours)** | Pseudo GT |

Figure 17: Comparison of image editing techniques for surface normal mapping. VISII and Textual Inversion yield unsatisfactory results, while InstructPix2Pix fails to interpret the task, resulting in near-original output.

We also provide a comparison with Bhattad et al. (2023a) in Tab. 5 and Fig. 18. This comparison is for the same 500 randomly generated images. Ours outperforms Bhattad et al. (2023a) significantly.

In addition, we show that directly applying SDEdit (Meng et al., 2021) will also fail to extract reasonable image intrinsics. We take the model from the SDv1-5 column in Fig.13 of the main paper and apply SDEdit. In Fig. 19, we show directly applying SDEdit results in severe artifacts, regardless of strength.

Table 5: Comparison of quality of normals extracted from StyleGAN Bhattad et al. (2023a).

| | Mean Error°↓ | Median Error°↓ | L1 × 100 ↓ |
|---|---|---|---|
| "StyleGAN knows" (Bhattad et al., 2023a) | 19.92 | 46.65 | 16.64 |
| LoRA-StyleGAN (Ours) | **13.24** | **23.55** | **10.92** |

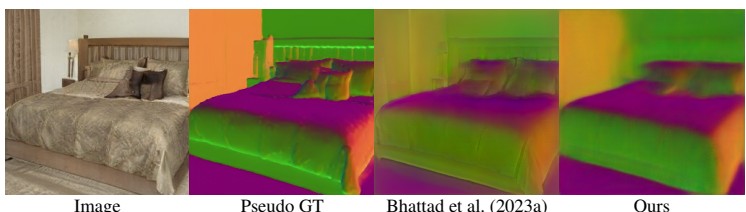

| Image | Pseudo GT | Bhattad et al. (2023a) | Ours |

Figure 18: Qualitative results of normals extracted from StyleGAN by Bhattad et al. (2023a) and Ours.

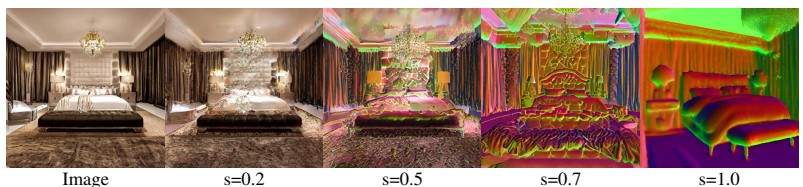

| Image | s=0.2 | s=0.5 | s=0.7 | s=1.0 |

Figure 19: We observe applying SDEdit method on the SDv1-5 model alone, without incorporating the additional input image latent encoding, fails to produce satisfactorily aligned and high-quality scene intrinsics. The reason for this might be the considerable domain shift that exists between RGB images and surface normal maps, which results in severe artifacts when using SDEdit. The variable "s" represents the strength of SDEdit.

## C  HYPER-PARAMETERS

In Table 6, we show the hyperparameters we use for each model.

## D  GENERATED IMAGES USED FOR QUANTITATIVE ANALYSIS

In Tab. 2 of the main paper, we report quantitative results on synthetic images. For Autoregressive models and GANs, we first randomly sample 500 noises and use them to generate 500 RGB images. The same 500 noises will then be used to generate intrinsics with our learned LoRAs loaded. For Stable Diffusion experiments (both single-step and multi-step), we use a single dataset with 1000 synthetic images with various prompts.

The pseudo GT are obtained by applying SOTA off-the-shelf models on the RGB images.

## E  ADDITIONAL QUALITATIVE RESULTS

In Fig. 20, we present more results for $SD_{AUG}$ and $SD1-5_{AUG}$. Fig. 21 shows extra results for models trained on FFHQ dataset. More examples of scene intrinsics extracted from StyleGAN-v2 trained on LSUN bedroom can be found in Fig. 22. In Fig. 23, we show results for SD-UNet (single-step) on generated images. Shown in Fig. 24 are extra results for StyleGAN-XL trained on ImageNet.

## F  RESULTS ON $1024^2$ SYNTHETIC IMAGES

Our multi-step $SD_{AUG}$ models, although trained exclusively on $512^2$ images from the DIODE dataset, demonstrate their robustness by successfully extracting intrinsic images from $1024^2$ high-resolution

Table 6: Hyper-parameters for each model. LR refers to the learning rate and BS refers to the batch size. Please note that the number of steps required to reach convergence reported above is for normal/depth. However, it is worth noting that albedo and shading tend to require significantly fewer steps to converge (usually half of normal/depth). Additionally, SD$_{\text{AUG}}$ (multi-step) and SD-UNet (single-step) are trained on real-world DIODE dataset, while the other models are trained on synthetic images within a specific domain. (Num. of params of VQGAN counts transformer + first stage models; Num. of params of SD$_{\text{AUG}}$ and SD-UNet counts VAE+UNet)

| Model | Dataset | Resolution | Rank | LR | BS | LoRA Params | Generator Params | Convergence Steps |
|---|---|---|---|---|---|---|---|---|
| VQGAN | FFHQ | 256 | 8 | 1e-03 | 1 | 0.13M | 873.9M | $\sim 4000$ |
| StyleGAN-v2 | FFHQ | 256 | 8 | 1e-03 | 1 | 0.14M | 24.8M | $\sim 4000$ |
| StyleGAN-v2 | LSUN Bedroom | 256 | 8 | 1e-03 | 1 | 0.14M | 24.8M | $\sim 4000$ |
| StyleGAN-XL | FFHQ | 256 | 8 | 1e-03 | 1 | 0.19M | 67.9M | $\sim 4000$ |
| StyleGAN-XL | ImageNet | 256 | 8 | 1e-03 | 1 | 0.19M | 67.9M | $\sim 4000$ |
| SD$_{\text{AUG}}$ (multi step) | Open | 512 | 8 | 1e-04 | 4 | 1.59M | 943.2M | $\sim 30000$ |
| SD-UNet (single step) | Open | 512 | 8 | 1e-04 | 4 | 1.59M | 943.2M | $\sim 15000$ |

synthetic images generated by Stable Diffusion XL (Podell et al., 2023), as shown across Figures 25 to 34.

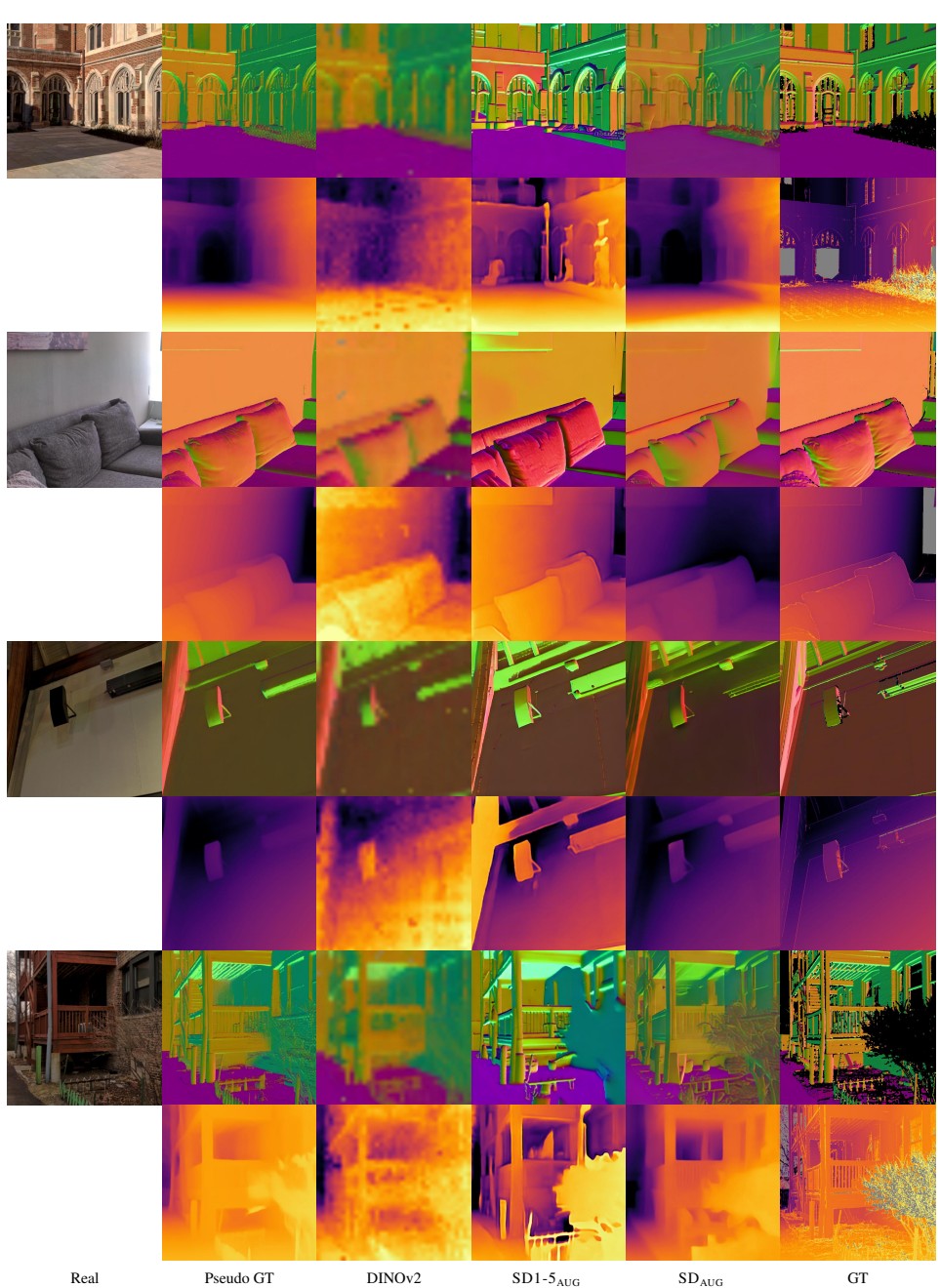

Real          Pseudo GT          DINOv2          SD1-5$_{AUG}$          SD$_{AUG}$          GT

Figure 20: Additional results after applying improved diffusion techniques with SD$_{AUG}$. SD$_{AUG}$ was found to significantly reduce color shift artifacts observed in SD1-5$_{AUG}$ during the extraction of detailed scene intrinsic results.

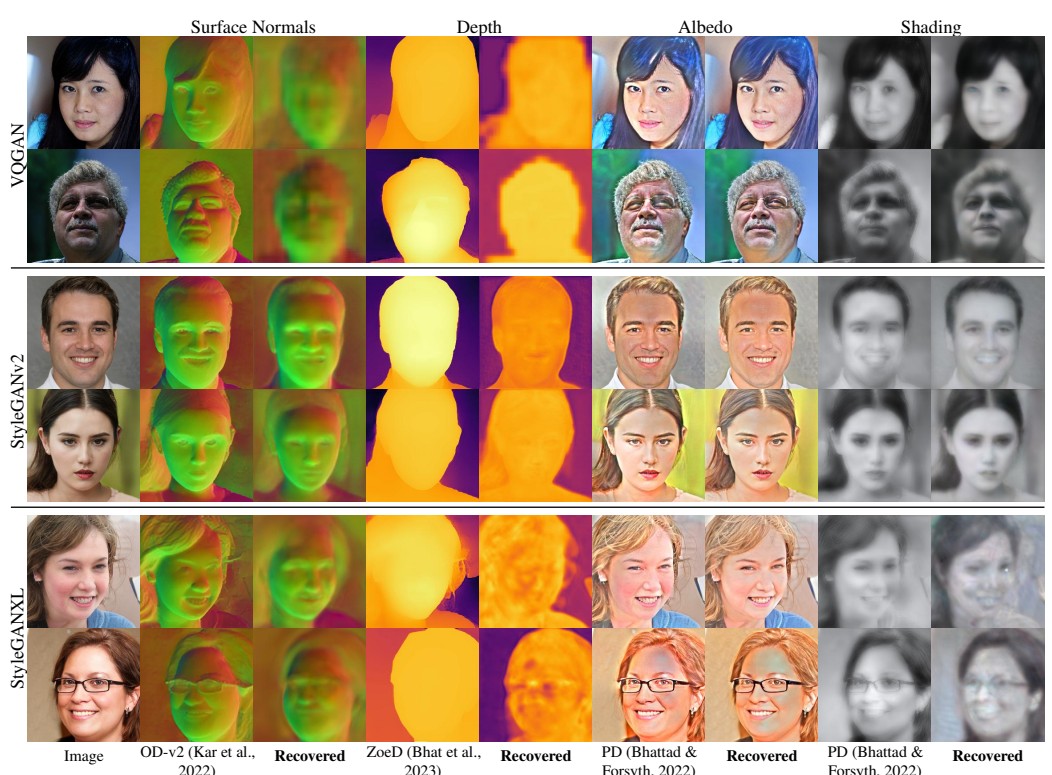

Figure 21: Additional results of scene intrinsics from different generators – VQGAN, StyleGAN-v2, and StyleGAN-XL – trained on FFHQ dataset.

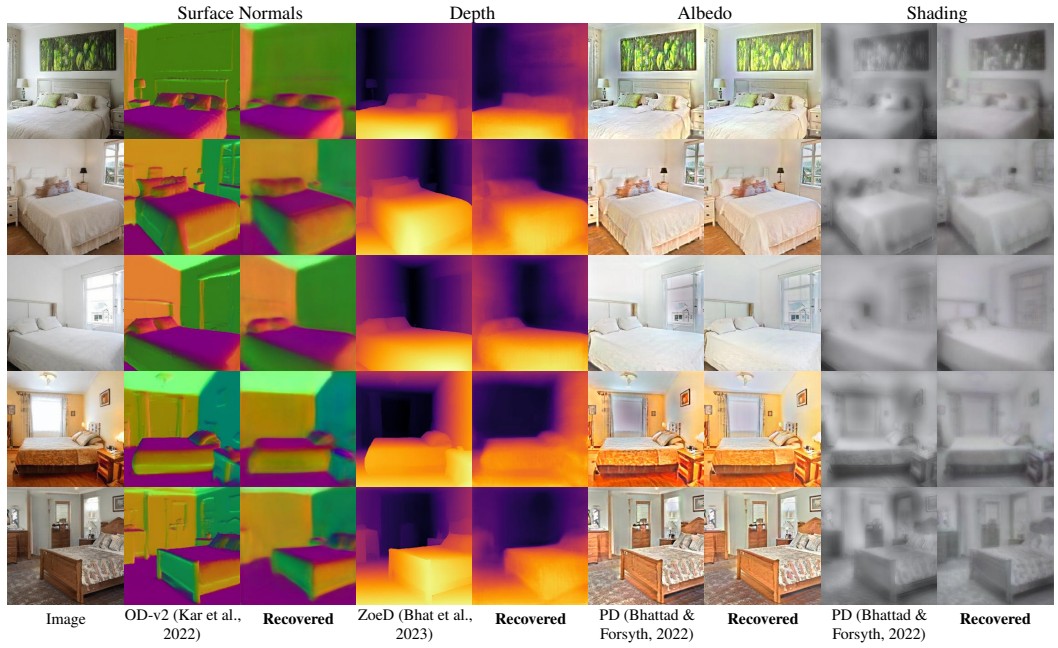

Figure 22: Additional results of scene intrinsics extraction from Stylegan-v2 trained on LSUN bedroom images.

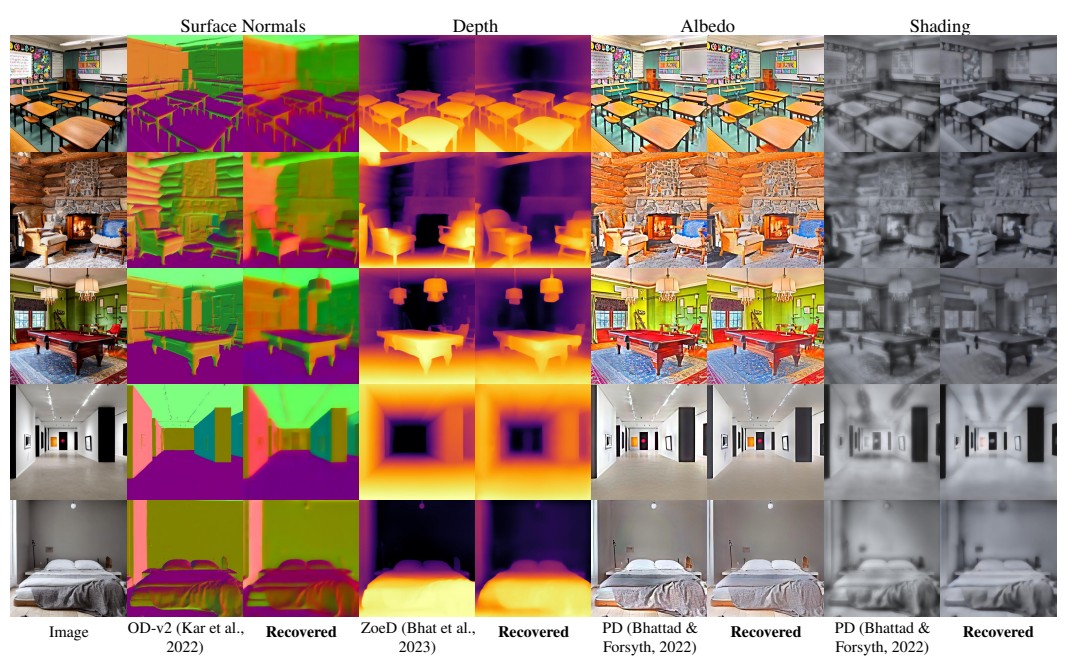

Figure 23: Additional results of scene intrinsics extraction from Stable Diffusion UNet (single-step).

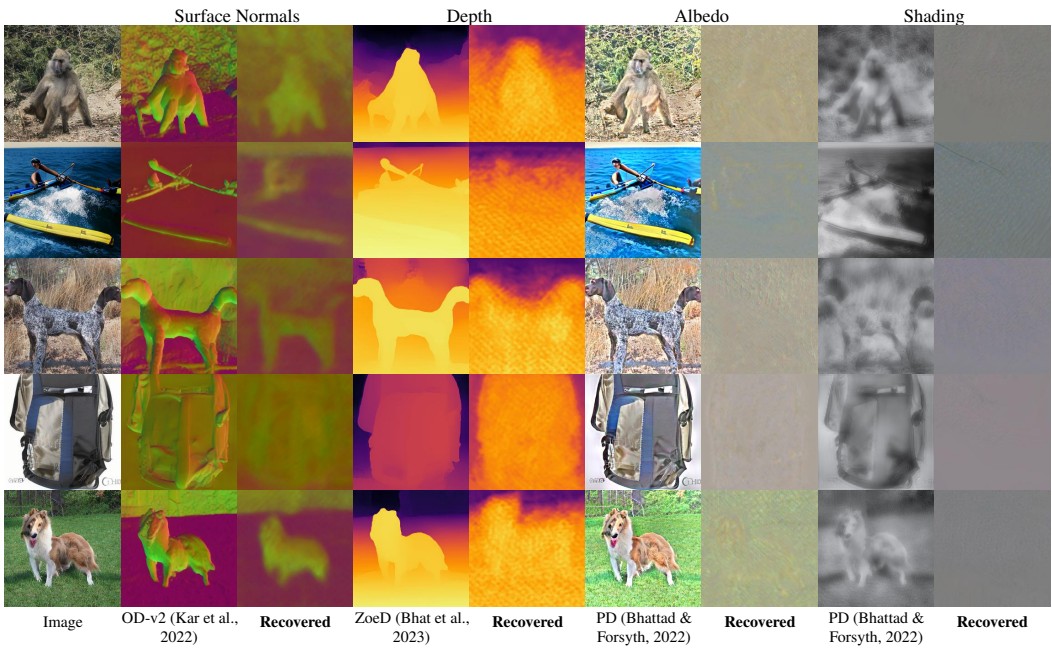

Figure 24: Additional results for StyleGAN-XL trained on ImageNet. StyleGAN-XL's inability to produce image intrinsics may be due to its inability to create high-quality plausible images.

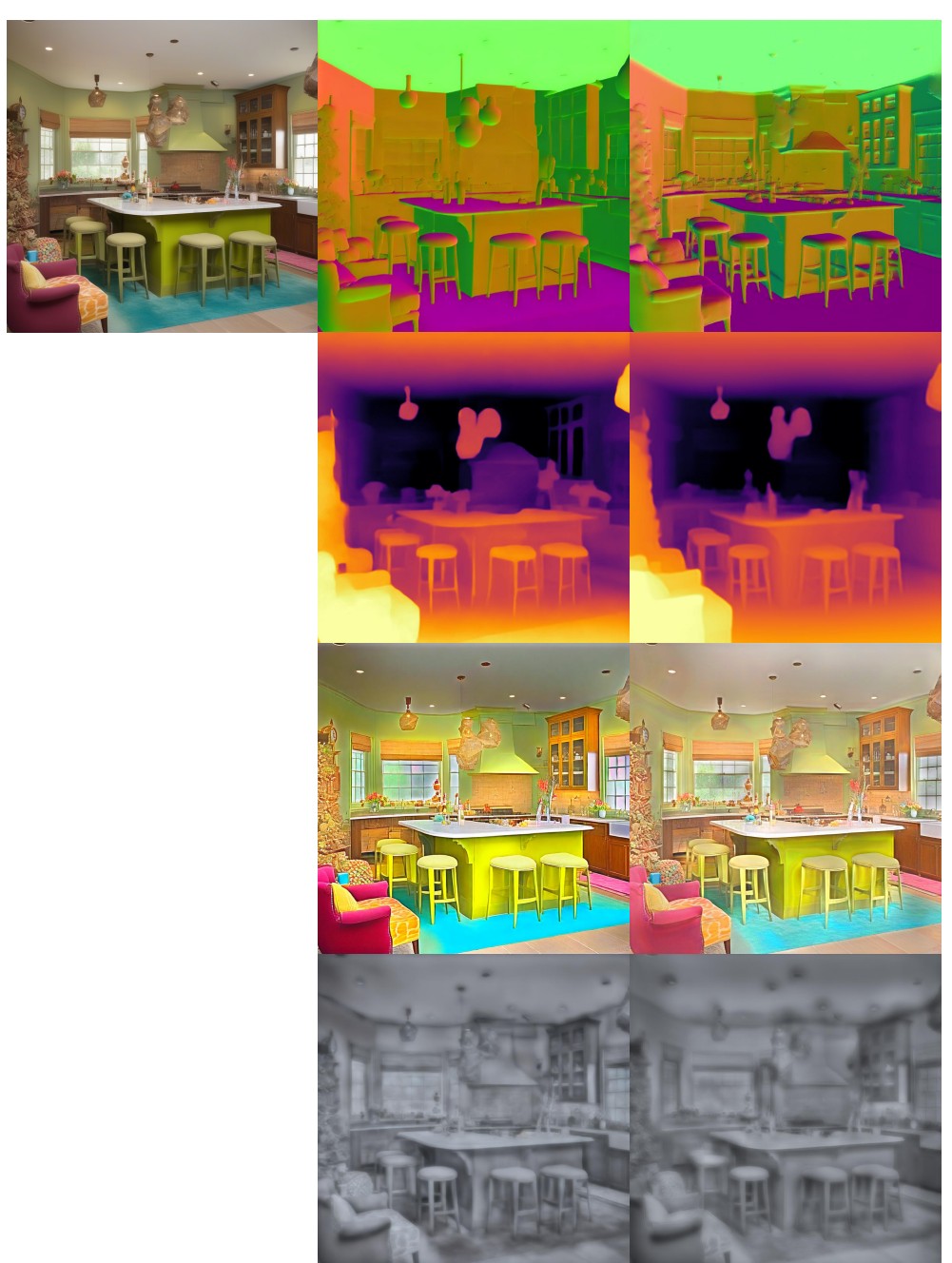

Figure 25: Results of SD$_{\text{AUG}}$ models applied on unseen $1024^2$ synthetic images. Left: original image; middle: ours; right: pseudo ground truth.

1242
1243
1244
1245
1246
1247
1248
1249
1250
1251
1252
1253
1254
1255
1256
1257
1258
1259
1260
1261
1262
1263
1264
1265
1266
1267
1268
1269
1270
1271
1272
1273
1274
1275
1276
1277
1278
1279
1280
1281
1282
1283
1284
1285
1286
1287
1288
1289
1290
1291
1292
1293
1294
1295

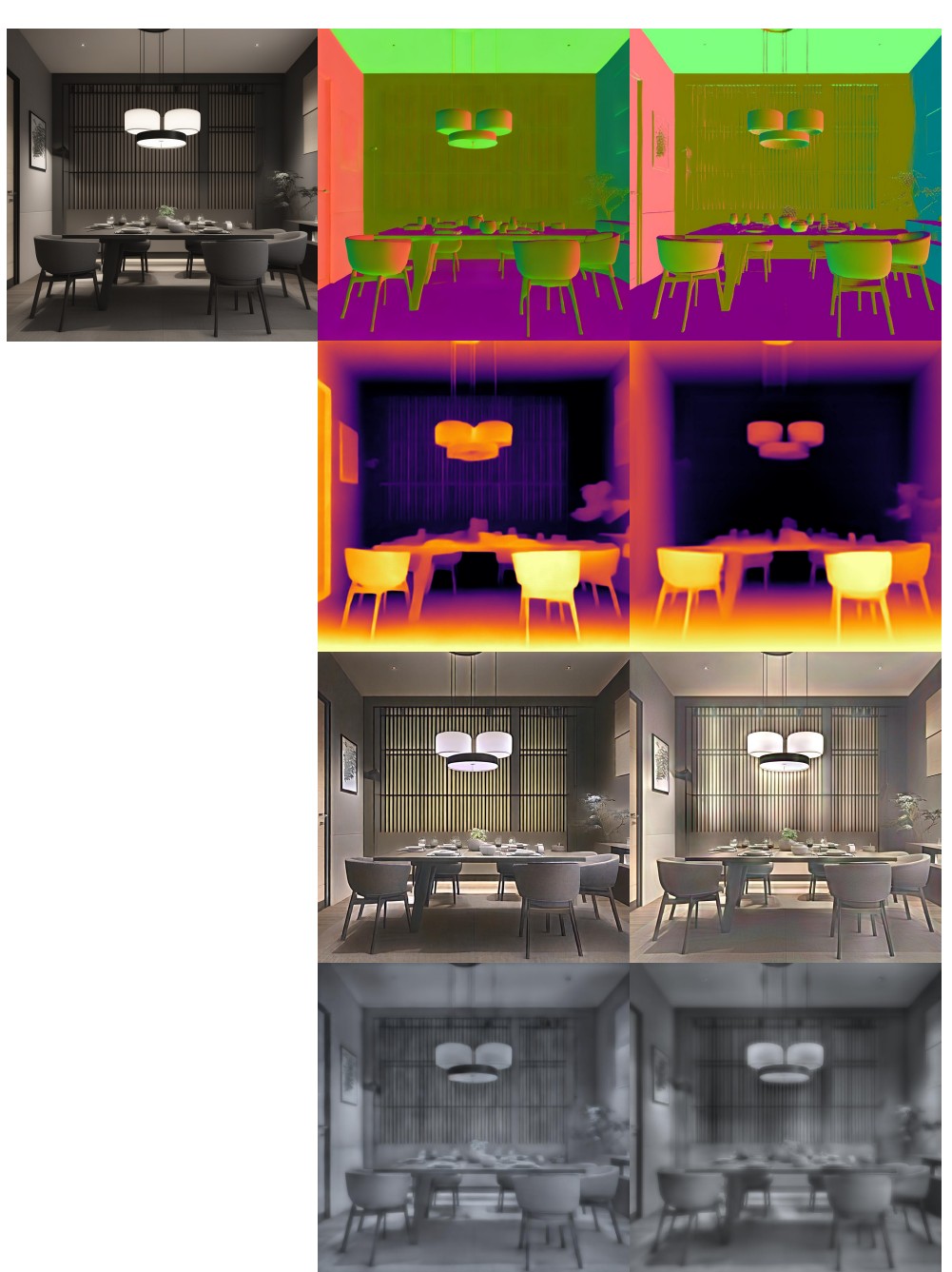

Figure 26: Cont. results of $SD_{AUG}$ models applied on unseen $1024^2$ synthetic images. Left: original image; middle: ours; right: pseudo ground truth.

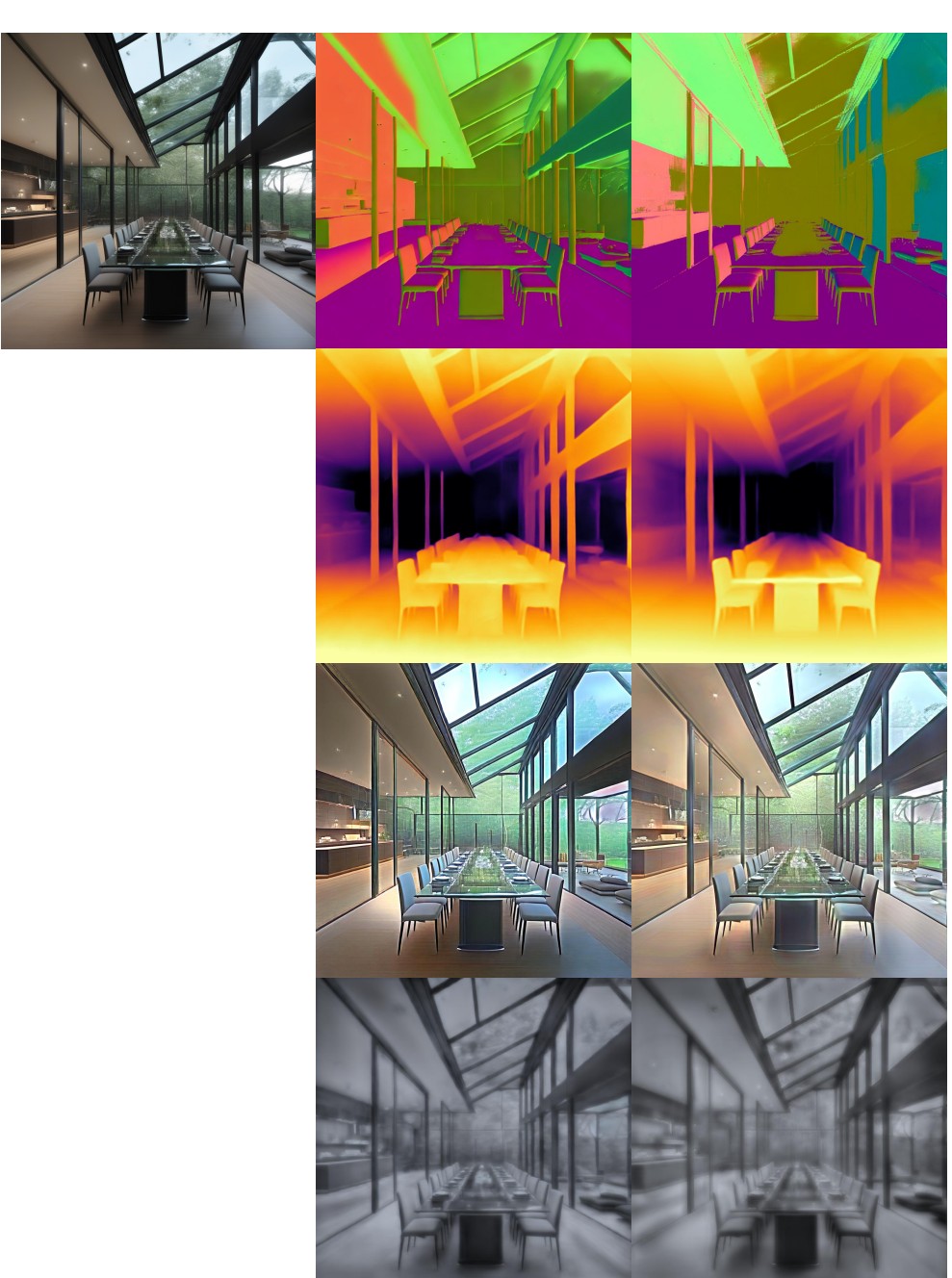

Figure 27: Cont. results of $\text{SD}_{\text{AUG}}$ models applied on unseen $1024^2$ synthetic images. Left: original image; middle: ours; right: pseudo ground truth.

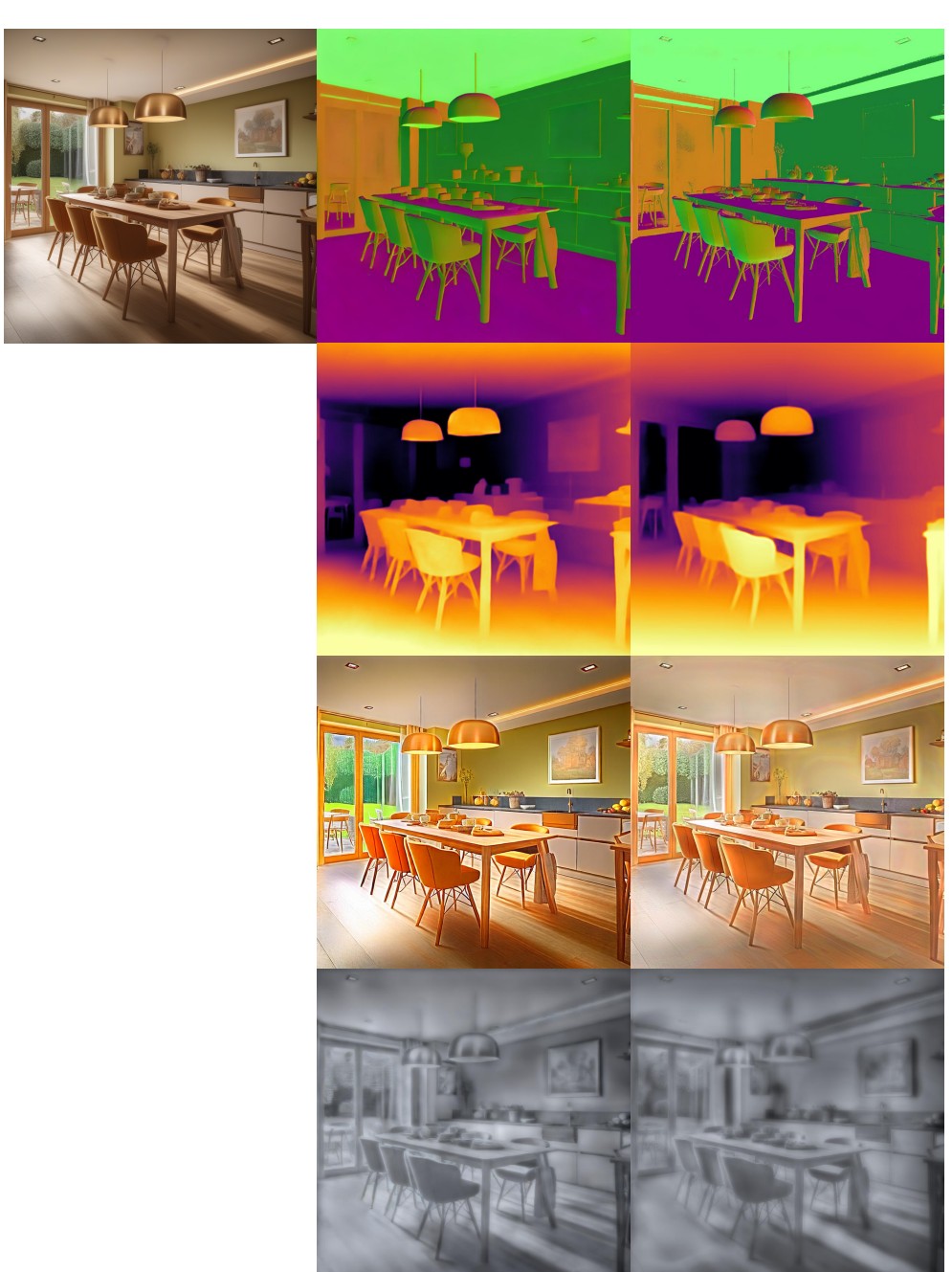

Figure 28: Cont. results of $SD_{AUG}$ models applied on unseen $1024^2$ synthetic images. Left: original image; middle: ours; right: pseudo ground truth.

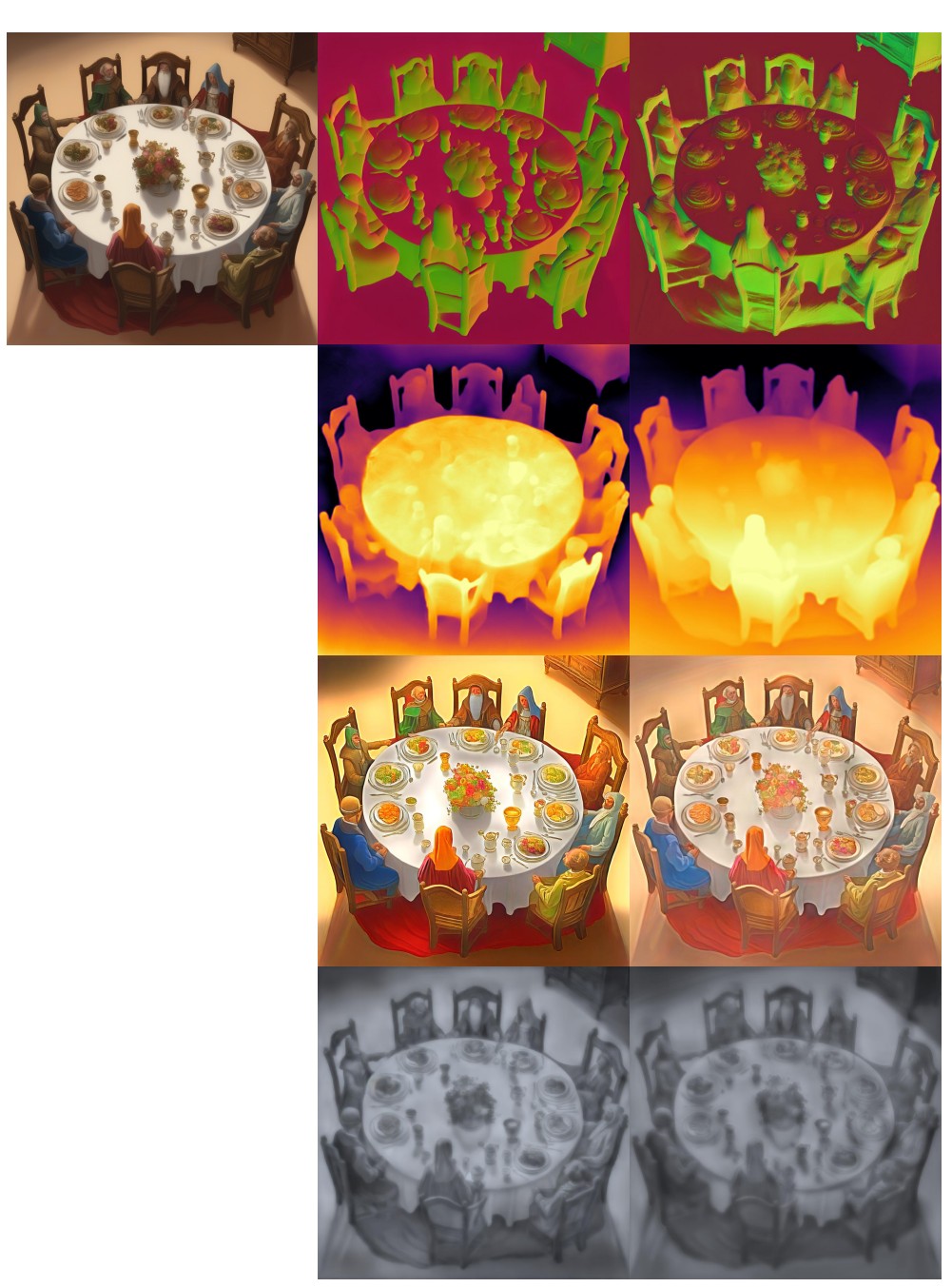

Figure 29: Cont. results of $SD_{AUG}$ models applied on unseen $1024^2$ synthetic images. Left: original image; middle: ours; right: pseudo ground truth.

1458
1459
1460
1461
1462
1463
1464
1465
1466
1467
1468
1469
1470
1471
1472
1473
1474
1475
1476
1477
1478
1479
1480
1481
1482
1483
1484
1485
1486
1487
1488
1489
1490
1491
1492
1493
1494
1495
1496
1497
1498
1499
1500
1501
1502
1503
1504
1505
1506
1507
1508
1509
1510
1511

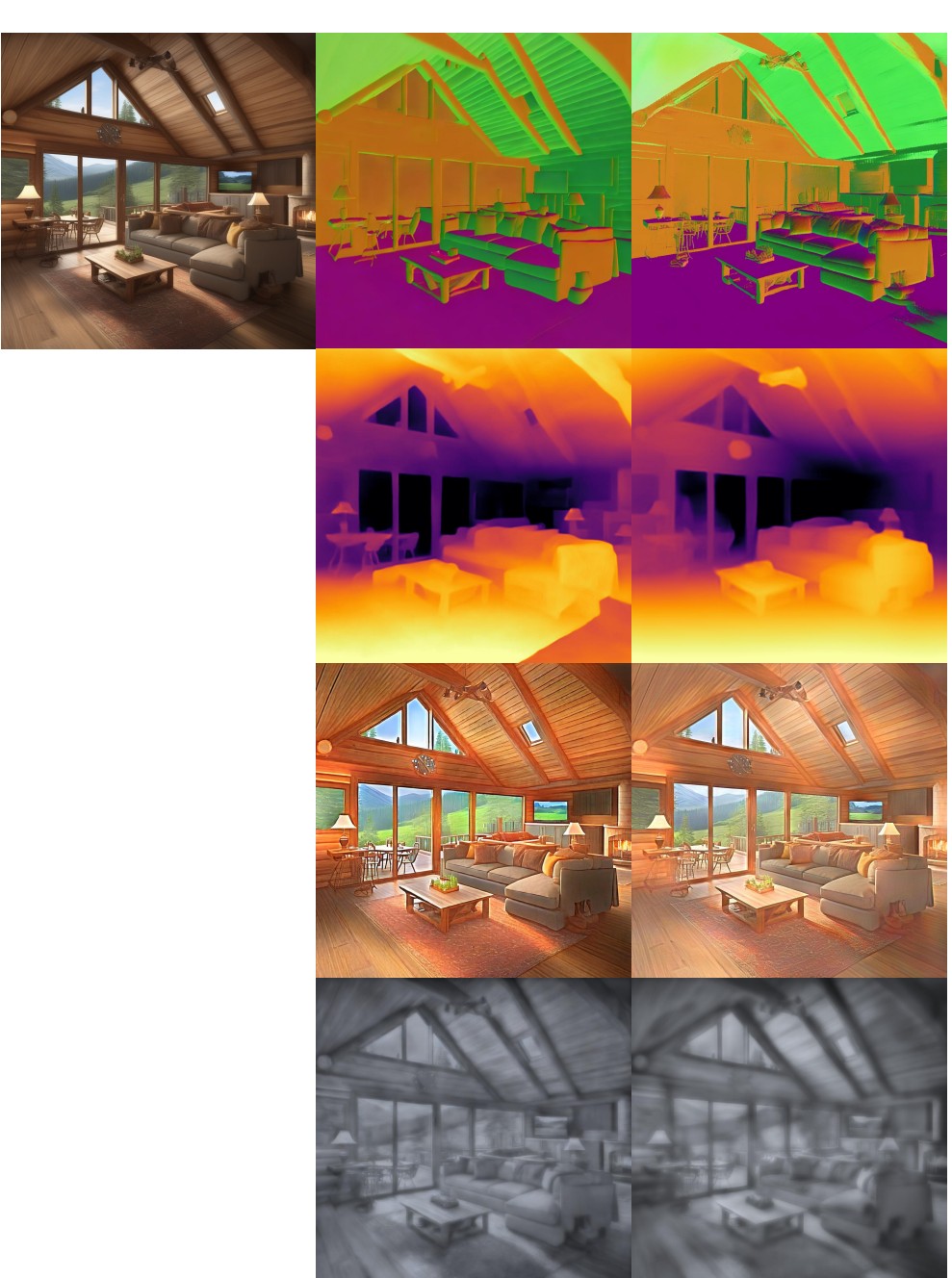

Figure 30: Cont. results of $\mathrm{SD_{AUG}}$ models applied on unseen $1024^2$ synthetic images. Left: original image; middle: ours; right: pseudo ground truth.

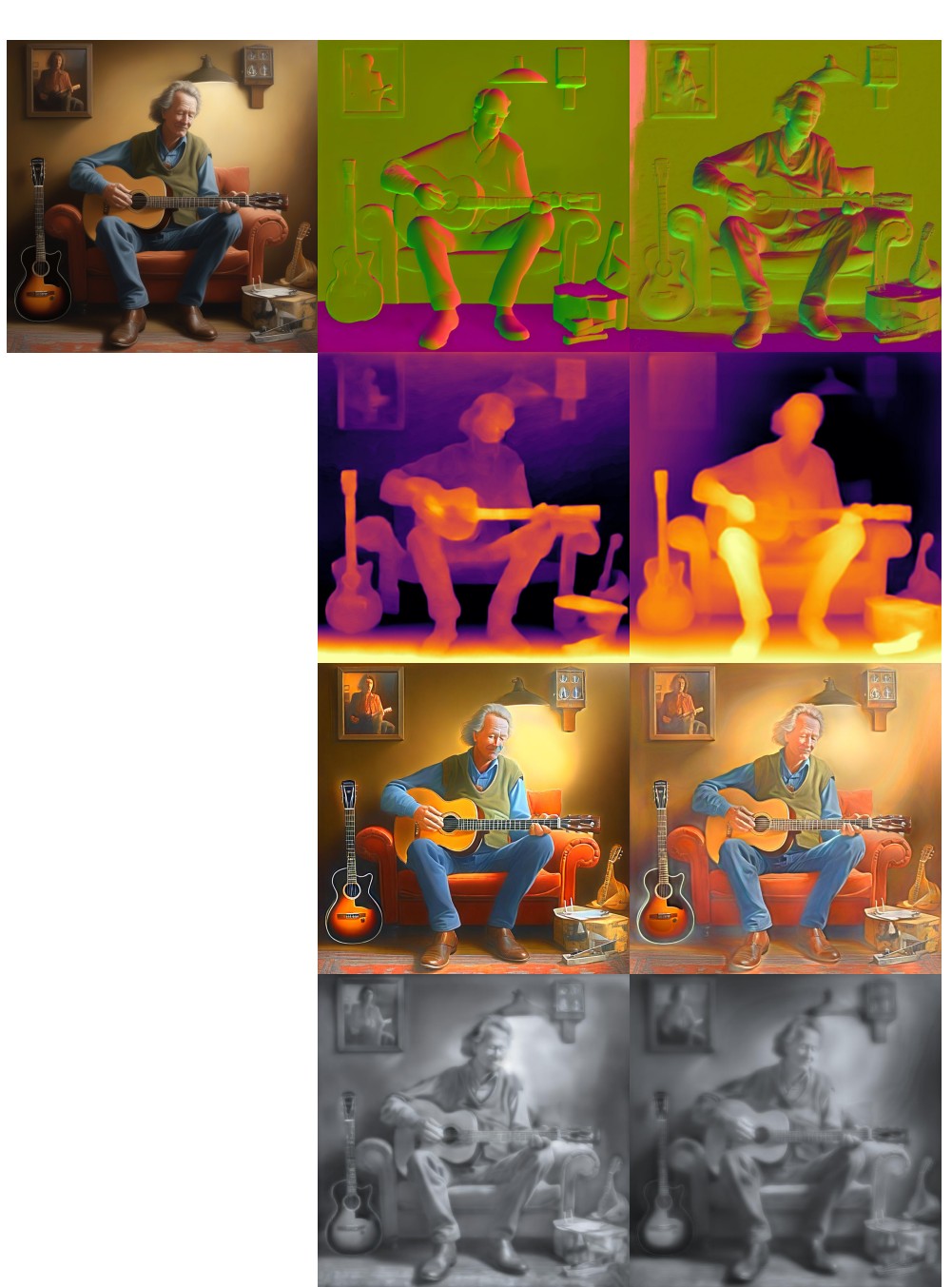

Figure 31: Cont. results of SD$_{\text{AUG}}$ models applied on unseen $1024^2$ synthetic images. Left: original image; middle: ours; right: pseudo ground truth.

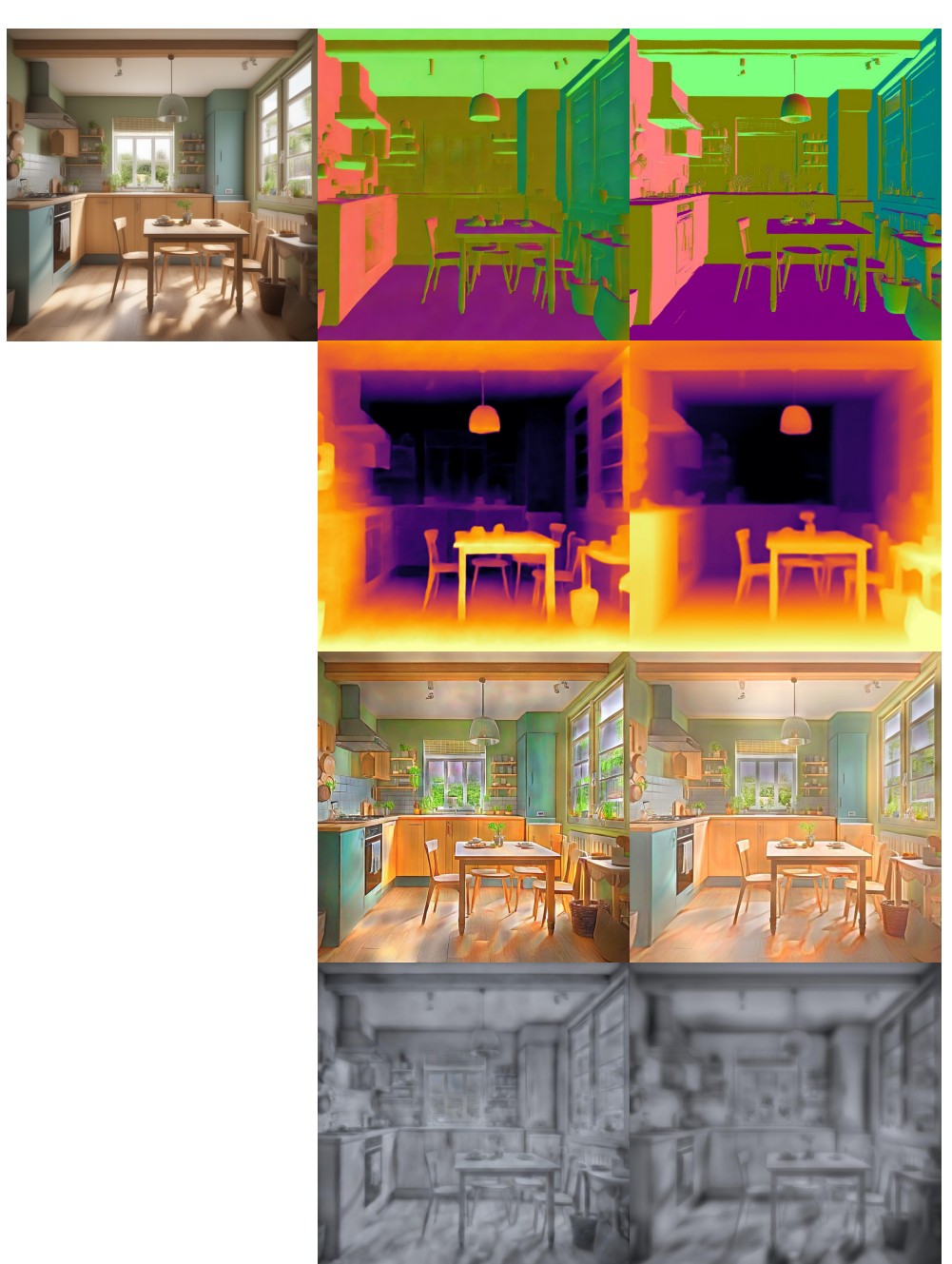

Figure 32: Cont. results of $SD_{AUG}$ models applied on unseen $1024^2$ synthetic images. Left: original image; middle: ours; right: pseudo ground truth.

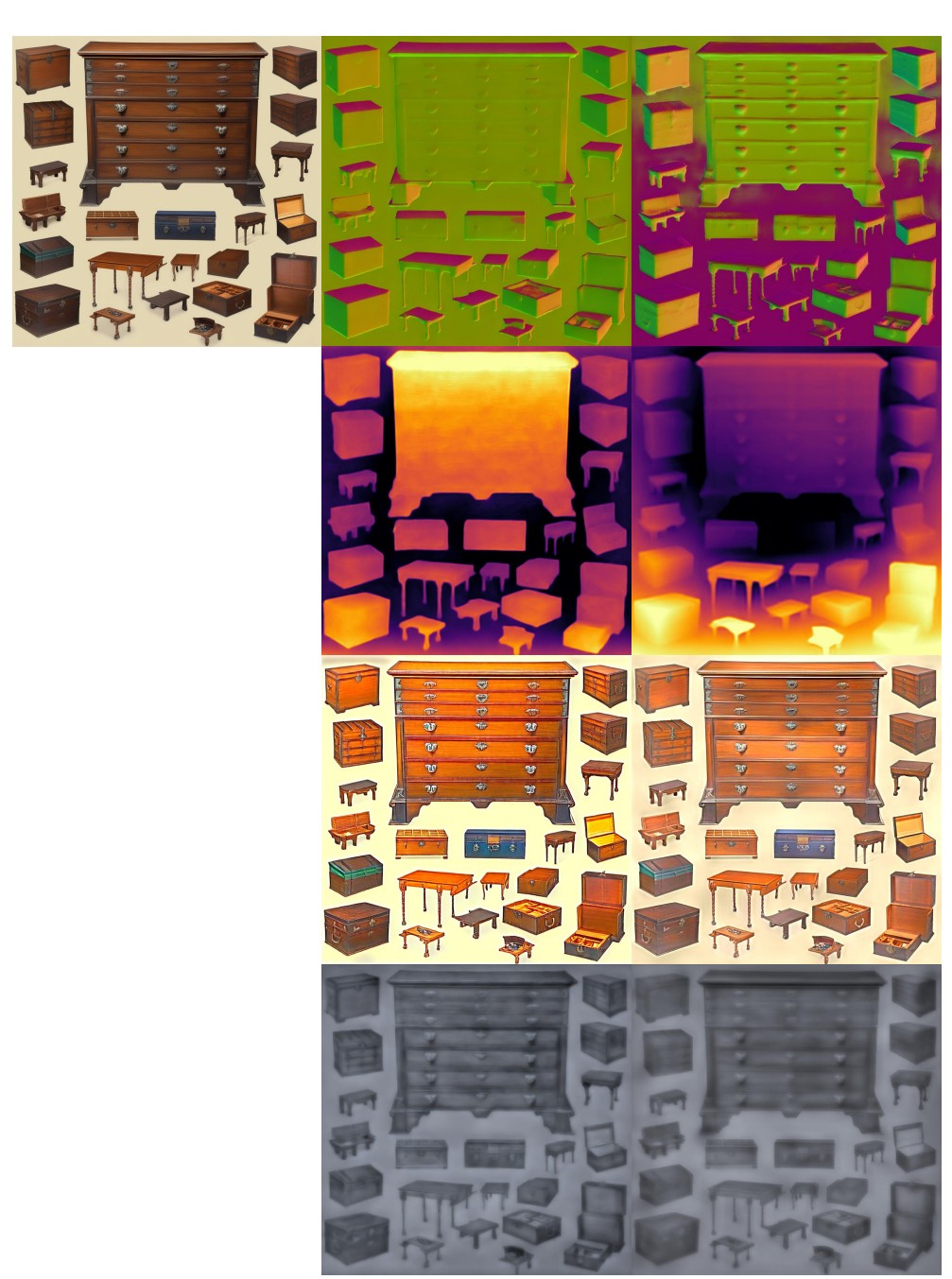

Figure 33: Cont. results of SD$_{\text{AUG}}$ models applied on unseen $1024^2$ synthetic images. Left: original image; middle: ours; right: pseudo ground truth.

1674
1675
1676
1677
1678
1679
1680
1681
1682
1683
1684
1685
1686
1687
1688
1689
1690
1691
1692
1693
1694
1695
1696
1697
1698
1699
1700
1701
1702
1703
1704
1705
1706
1707
1708
1709
1710
1711
1712
1713
1714
1715
1716
1717
1718
1719
1720
1721
1722
1723
1724
1725
1726
1727

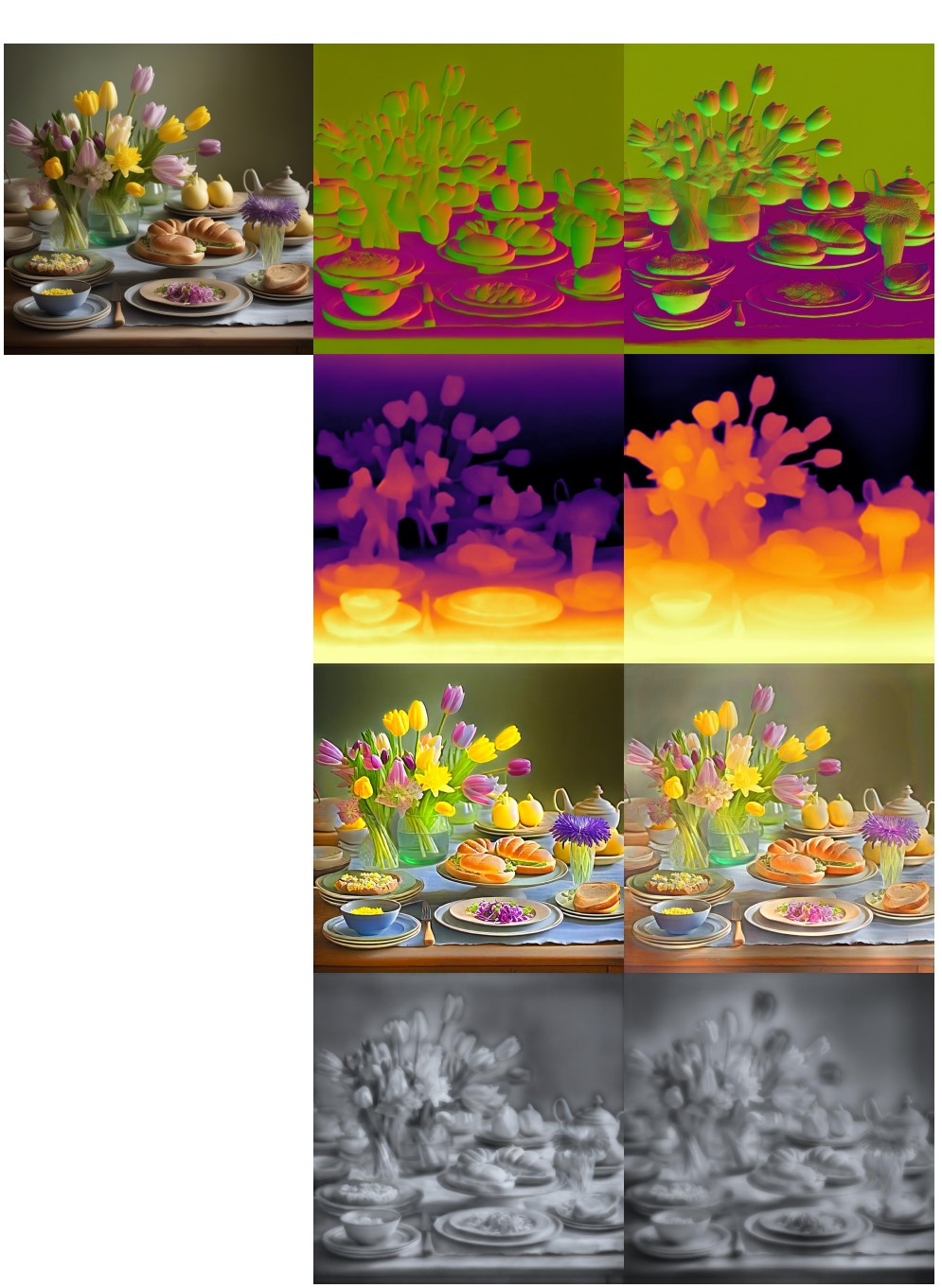

Figure 34: Cont. results of $SD_{AUG}$ models applied on unseen $1024^2$ synthetic images. Left: original image; middle: ours; right: pseudo ground truth.

