# OpenReview forum: "Generative Models: What Do They Know? Do They Know Things? Let's Find Out!"
_ICLR.cc/2025/Conference — Submitted to ICLR 2025_

### Official Review · Reviewer_iQ6x · 2024-11-01

**Soundness:** 3
**Presentation:** 4
**Contribution:** 3
**Rating:** 6
**Confidence:** 3

**Summary:**

This paper proposes to use pretrained generative models extended with trainable LoRA layers as image intrinsic predictors. The proposed approach aims to learn effective intrinsic extractors with as few LoRA parameters and training samples as possible. Extensive experiments are conducted to primarily show that (1) with less LoRA parameters and data samples than state-of-the-art approaches, extracting intrinsic images is still possible (2) it is the prior knowledge in pretrained generative models that helps extract intrinsic image, or in other words, generative models do encode useful intrinsic knowledge though it is not clear how such intrinsic knowledge are used during generation.

**Strengths:**

1. The paper is well-written and easy-to-follow

2. The approach is straightforward and effective by using LoRA to adapt pretrained generative models for downstream tasks beyond image generation. The predicted intrinsic results look decent.

3. By using LoRA instead of retraining or finetuning the whole model, the approach requires less data and parameters to perform the image intrinsic estimation task. As an experimental work, the paper mostly supports the claims therein with extensive experiments.

**Weaknesses:**

1. Fig 8 and Fig 9 show that there is a performance peak as the number of LoRA params or training samples vary. Why there is a peak? For  the case of fixed number of training samples, with more LoRA params than Rank 8, the performance degrades. Is this due to underfitting that there is no enough data to train larger LoRA well enough? Similarly, for the case of fixed number of  LoRA params, is this due to overfitting that more data overfits a relative small LoRA?

2. A relevant question to 1.  If you increase the number of LoRA and the number of training data at the same time avoiding overfitting or underfitting, can you outperform existing approaches eventually as shown in Table 3? Furthermore, the paper shows qualitative results for all four intrinsics Surface Normal, Depth, Albedo and Shading throughout the paper, but shows quantitative results on only Surface Normal, Depth in Table 3. Is there is any reason for this practice?

3. The paper overclaims a bit about "minimal" requirements on parameter updates and data. Why do you say minimal and in what sense it is minimal? Thought the approach is able to work with fewer params and data , the quantitative performance doesn't outperform existing approaches as shown in Table 3.

4. Typo Table 3 caption, "intrinsicsacross"->"intrinsics across"

**Questions:**

To AC: I can clearly understand the task, the proposed approach and the experiments in the paper. Overall, I find the paper supports the claims with properly designed experiments, but I cannot evaluate the importance and novelty of the paper at the moment since I don't follow literature on this intrinsic estimation topic. I would evaluate these two aspects later by referring to opinions from other reviewers. So please use this review with proper weight when make decisions.

---

> ### Author Response · Authors · 2024-11-22
> **Authors' Response to Reviewer iQ6x**
>
> 1. For the case of fixed training samples, ranks higher than 8 are probably overfitting to the limited number of training samples. For the case of fixed ranks, we observe that during training, the norm of LoRA with larger data fluctuates a lot more than smaller data. This could imply the complexity of the larger dataset makes it harder for adapters to converge.
>
> 2. It could be possible to outperform numbers in Tab.3. However, our goal is to use as little data and as few learnable parameters as possible. Please also see our general response on the scope of the paper. The main focus of this paper is not to achieve SOTA on intrinsics prediction; otherwise, we can always fine-tune all the weights or directly learn a complex decoder head.
>
>     We did not have quantitative results of Albedo and Shading for Tab.3 because Tab. 3 is for results on the DIODE dataset. DIODE dataset only provides Normals and Depths, not the Albedo and Shading. In fact, ground truth Albedo and Shading is extremely hard, if not impossible, to obtain for real data. Please see the recent work by Zhang et al NeurIPS 2024, which discusses the ill-posedness of intrinsic image decomposition.
>
>     **Reference:**
>     - Zhang X, Gao W, Jain S, Maire M, Forsyth D, Bhattad A. Latent Intrinsics Emerge from Training to Relight. arXiv preprint arXiv:2405.21074. 2024.
>
> 3. We apologize for the confusion. The word “minimal” only means “very small or slight”. We didn't and don't want to claim our approach uses the least possible amount of data among all existing works. We do realize now the word can create confusion and we have already updated the PDF to replace the word “minimal” with “light” and “small”.
>
> 4. We would like to thank the reviewer for pointing it out. We have already fixed this typo and updated the PDF.

---

> ### Author Response · Authors · 2024-11-25
> **Looking Forward to Your Feedback**
>
> We sincerely appreciate the time and effort you put into reviewing our paper. Every question you raised has been carefully reviewed and addressed, with the changes reflected both in our detailed responses and the revised PDF. As the Reviewer-Author discussion phase ending soon, we are eager to hear any further comments you might have. If you have any additional questions, we would be more than happy to provide thorough explanations.

---

> ### Author Response · Authors · 2024-11-30
> **Follow-up on Responses to Your Review**
>
> Dear Reviewer iQ6x,
>
> Following our previous response to your comments, we wanted to check if you have any additional questions or require any clarifications. We would greatly appreciate knowing if our responses have adequately addressed your concerns, or if there are any other aspects of the paper that need further explanation.Thank you for your time and valuable feedback.
>
> Sincerely,
>
> Authors of Paper 4684

---

> ### Author Response · Authors · 2024-12-03
> **Consideration for Score Adjustment Following Revisions**
>
> Thank you for your insightful comments and feedback. Your suggestions have helped us improve the paper. Every question you raised has been carefully reviewed and addressed, with the changes reflected both in our detailed responses and the revised PDF.
>
> If your view of the paper has evolved, we kindly ask that you consider updating your score.
>
> Thank you very much for your time.

---

### Official Review · Reviewer_RfeE · 2024-11-02

**Soundness:** 2
**Presentation:** 3
**Contribution:** 3
**Rating:** 6
**Confidence:** 4

**Summary:**

The paper investigates whether current generative models possess a genuine understanding of the world they're generating, rather than just statistical pattern matching. The authors propose using the generation of image intrinsics (such as depth maps, normal maps, etc.) as a proxy for evaluating the models' physical understanding and explainability. They demonstrate that a simple adaptation method (LoRA) can effectively recover these intrinsic properties and conduct extensive analyses to support their hypothesis about models' world understanding.

**Strengths:**

- The work raises fundamental questions about the nature of internal representations in generative models and their relationship to generation quality, contributing to our understanding of these architectures.
- The research demonstrates that world representations in powerful generative models are more concrete and accessible than previously assumed, establishing important connections to explainable AI.
- The authors present compelling insights about the minimal data requirements for intrinsics recovery, suggesting that the underlying knowledge is already embedded in the model's parameters.
- The methodology is straightforward yet effective, making the findings easily reproducible and applicable to various generative models.

**Weaknesses:**

- The paper appears to have originated as a study specifically focused on diffusion models, with other generative architectures added later to broaden its scope. While Section 4's experiments and Section 5's generation quality refinements primarily use Stable Diffusion, validating these findings across different architectures would strengthen the paper's claim of developing a general approach.
- The experiments in Section 4.2 focus solely on normal map recovery when determining optimal rank and dataset size. The authors should clarify whether these optimal settings generalize to other intrinsics or if each intrinsic property requires specific configurations.
- The paper would benefit from a deeper analysis of why LoRA is particularly effective for intrinsic recovery. While Line 102 suggests that "intrinsic information is distributed throughout the network," this insight isn't fully developed in Section 4.4 or elsewhere. Understanding this mechanism could also inform the development of future recovery methods.

The paper presents strong empirical findings, though deeper analysis is needed to enhance its impact on interpretability and explainable AI research.

**Questions:**

- Related to the last weakness, Section 4.4 identifies LoRA as the superior method for intrinsic image recovery, but the underlying reasons remain unclear. Could the authors provide insights into its advantages over alternative methods?
- The authors demonstrate a positive correlation between image intrinsics and generation quality. Could they elaborate on the nature of this relationship? Specifically, does better intrinsic understanding lead to improved generation quality, vice versa, or is there a bidirectional relationship between these aspects?

---

> ### Author Response · Authors · 2024-11-22
> **Authors' Response to Reviewer RfeE**
>
> ### Weaknesses Section
> - &nbsp;&nbsp;&nbsp;&nbsp;&nbsp;&nbsp;Thanks for the question. Our study of other generative models is not an afterthought. Please see our general response on scope and novelty as well as our Introduction section. Our paper is driven by curiosity to understand why generative models are so successful. Furthermore,  our paper has already presented many experiments for all generative architectures in question. In Fig. 4 (SG-XL, SGv2 and VQGAN on FFHQ), Fig.5 (SGv2 on LSUN bedroom), Fig.6 (SG-XL on ImageNet), Fig.7 (SD) and Tab. 2 in Sec.4, we report both quantitative and qualitative results for all generative models we mentioned in the paper.
>
>     &nbsp;&nbsp;&nbsp;&nbsp;&nbsp;&nbsp;The experiments/ablations that only contain diffusion models are for *real* data. As we explained in Line 371-372, only diffusion models allow inputting of real images. There is no direct way to conduct the same experiments with GANs or autoregressive models, which only accept noise vectors as input. Therefore, it may appear to have a more substantive evaluation of Diffusion Models.
>
>     &nbsp;&nbsp;&nbsp;&nbsp;&nbsp;&nbsp;Similarly, Sec.5 is an extension of our approach to multi-step inference. It is only meaningful to diffusion models. GANs and autoregressive models only support single-step inference.
>
> - &nbsp;&nbsp;&nbsp;&nbsp;&nbsp;&nbsp;We only show results on normal maps due to compute resource and page length limitations. The optimal settings generalize to other intrinsics and we use these settings across all intrinsics for diffusion models in the rest of the paper.
>
>     &nbsp;&nbsp;&nbsp;&nbsp;&nbsp;&nbsp;As for GANs and Autoregressive models, since they cannot accept real images, there is no notion of dataset size. New images are sampled for each training iteration. The optimal rank, however, generalizes to all types of architectures, including diffusion models, GANs and Autoregressive models.
>
>     &nbsp;&nbsp;&nbsp;&nbsp;&nbsp;&nbsp;We have already updated the PDF to emphasize the generality of the settings at Line 423.
>
> - &nbsp;&nbsp;&nbsp;&nbsp;&nbsp;&nbsp;We are sorry for the confusion. Because of the page limit, we include the analysis in Appendix B and Fig. 16. We will clearly state this reference in the final manuscript. Please also refer to our response to Reviewer w4yW’s question **[W2.3]**
>
> ### Questions Section
>
> - &nbsp;&nbsp;&nbsp;&nbsp;&nbsp;&nbsp;LoRA strikes the balance between parameter efficiency and data efficiency. Linear probing is parameter efficient, but the low resolution of the latent space of Stable Diffusion restricts its accuracy, and also linear probing does not account for the fact that the information may be spread out at different layers in the network. Please refer to our answer to **Reviewer 1965’s 3rd question** for a more detailed discussion. Full fine-tuning, on the other hand, can easily overfit to the training data and has the risk of completely destroying the learned knowledge.
> - &nbsp;&nbsp;&nbsp;&nbsp;&nbsp;&nbsp; This is an excellent question that we are also very eager to investigate. We believe the relationship is bidirectional as we did not see a “sudden” emergence of one aspect while the other one stays smooth/unchanged. However, this is a deep topic that we hope to investigate in future work.

---

> > ### Comment · Reviewer_RfeE · 2024-11-28
> >
> > Thank you for the detailed responses to my questions and concerns. I found your clarifications helpful and appreciate the additional references to sections of the paper and appendix.
> >
> > **W1 and W2**: Your explanations adequately address my concerns regarding the broader applicability of your approach across generative models. I appreciate the detailed clarification regarding the challenges specific to GANs and autoregressive models. While I understand that direct analysis of real images isn't feasible with these architectures, inversion techniques could be employed to map real images to their latent space. This could potentially allow for similar analyses to those conducted on diffusion models.
> >
> > **W3**: Thank you for pointing out the analysis in Appendix B. I found it insightful and appreciated the comparisons of editing methods. However, I believe this analysis deserves a more central focus in the main paper to better explain why intrinsic recovery works with LoRA fine-tuning. Additionally, the inclusion of a clearer conclusion from this analysis would enhance the paper’s interpretability. For example, methodologies like [1] and [2] offer layer-specific analyses that could be adapted here. By selectively deactivating LoRAs and observing the impact on intrinsic recovery, it would be possible to identify where in the network intrinsic knowledge is most concentrated. This deeper analysis would not only clarify the mechanism, but also align well with the paper's broader goals of improving explainability.
> >
> > **Q1**: Thank you for the clarification regarding why LoRA is particularly effective for intrinsic recovery. I find the observation that “information is spread across different layers in the network” to be a crucial insight. Including this in the main paper beyond the Introduction would improve the intuitive understanding of LoRA’s advantages and enhance the accessibility of your findings.
> >
> > **Q2**: I look forward to it. Concurrent works, such as [3], also hint toward similar relationships, reinforcing the importance of this line of inquiry.
> >
> > Overall, I believe the paper raises important questions and provides meaningful contributions to understanding the internal representations of generative models. While the authors' responses address the primary concerns, I see room for further improvements by incorporating more intuitive explanations and centralizing key analyses currently relegated to the appendix.
> >
> > In recognition of the paper’s merits and the thoughtful responses provided, I have updated my score to a 6. I encourage the authors to continue enhancing the paper by integrating these suggestions, as they would further solidify its impact in the fields of explainable AI and generative modeling.
> >
> > [1] InstantStyle: Free Lunch towards Style-Preserving in Text-to-Image Generation
> > [2] P+: Extended Textual Conditioning in Text-to-Image Generation
> > [3] Representation Alignment for Generation: Training Diffusion Transformers Is Easier Than You Think

---

> > > ### Author Response · Authors · 2024-11-30
> > >
> > > We greatly appreciate your recognition of the contributions of our work and your decision to increase the score.
> > >
> > > We will incorporate your recommendations in the final manuscript. Specifically, we will move the analysis in Appendix B to the main paper and include a clearer explanation. Additionally, we will consider conducting a layer-by-layer analysis, as you suggested, for the next step.
> > >
> > > We will continue to strengthen our paper throughout the review process.
> > >
> > > Thank you again for your detailed feedback and valuable suggestions. Should you have any further suggestions for improvement, please let us know. Your feedback has been extremely valuable in improving our work.

---

> ### Author Response · Authors · 2024-11-25
> **Looking Forword to Your Feedback**
>
> We sincerely appreciate the time and effort you put into reviewing our paper. Every question you raised has been carefully reviewed and addressed, with the changes reflected both in our detailed responses and the revised PDF. As the Reviewer-Author discussion phase ending soon, we are eager to hear any further comments you might have. If you have any additional questions, we would be more than happy to provide thorough explanations.

---

### Official Review · Reviewer_w4yW · 2024-11-02

**Soundness:** 3
**Presentation:** 3
**Contribution:** 2
**Rating:** 6
**Confidence:** 4

**Summary:**

This paper investigates the intrinsic information (e.g., normal, depth, albedo, and shading) encoded in various generative models, including GANs, autoregressive models, and diffusion models. Key findings include (1) Intrinsic information is a byproduct of training generative image models; (2) Low-Rank Adaptation (LoRA) is a generic technique to study the intrinsic information encoded, better than other approaches such as linear probing and fine-tuning; (3) The quality of a generative model and accuracy of its recovered intrinsics (e.g., depth prediction accuracy) are positively correlated. To recover each intrinsic property, a structured prediction module (e.g., depth prediction) has been used to provide pseudo groundtruth during the LoRA optimization. Experiments have been conducted mainly on depth and normal prediction.

**Strengths:**

- [S1: Originality] Though the proposed method is very standard, the paper presented a generic approach and demonstrated that the approach recovers intrinsic information across diverse generative models, including GANs, autoregressive models, and diffusion models.

- [S2: Clarity] The motivation, methodology, and results are clearly articulated, enabling a good understanding of the research contributions.

**Weaknesses:**

- [W1: Deeper understanding] While the paper establishes what intrinsic knowledge is encoded, it doesn't delve into how generative models utilize this knowledge during image generation.
  - [W1.1] For example, the reviewer would like to understand if intrinsic property emerges with a half-trained generative model (or at different training epochs).

- [W2] The reviewer find that most intrinsic property do not contain high-frequency details (e.g., sharp edges in the depth prediction). This could be the shortcoming of using LoRA fine-tuning in the design. Although multi-step diffusion inference has been added in Section 5, the question still remains if a single-step approach is sufficient to recover high-fidelity intrinsic information.
  - [W2.1] How does multi-step inference apply to other generative models such as StyleGAN?
  - [W2.2] What’s the motivation of applying LoRA to attention layers of a diffusion model?
  - [W2.3] The ablation studies on applying LoRA to different modules (Appendix B) is interesting. It seems to suggest that LoRA is not successful when applied to up or down blocks. What’s the insight behind the discovery?

- [W3] The reviewer does not fully understand the claim that StyleGAN-XL trained on ImageNet is an exception (Line 365). For example, StyleGAN-XL achieved a much lower FID (Line 452) compared to other generative models, which seems to contradict with the claim that StyleGAN-XL’s limited ability to generate realistic images (Line 366). Please clarify this point in the rebuttal.

- [W4] Specific to the comparison between SD-1.X and SD-2.1,  how much of the performance difference can be attributed to the improved encoder-decoder? Is it possible to recover the intrinsic property by applying LoRA fine-tuning on encoder-decoder alone?

**Questions:**

Please address the comments in the weakness section.

---

> ### Author Response · Authors · 2024-11-22
> **Authors' Response to Reviewer w4yW**
>
> [W1] We also discussed this point in the Limitation section at Line 535-536. Decoding how deep neural networks utilize knowledge internally is still an ongoing challenge for the community. It can be the topic of our follow-up work. The main goals of this paper are the four questions we raised in the abstract.
>
> - [W1.1] Our discussion in Sec. 4.3 is highly related to this question. SDv1-1, SDv1-2 and SDv1-5 are with exactly the same architecture, each being a continued training checkpoint of the previous. As we report in Fig.10, with more training epochs/better generative ability, the extracted intrinsics are also more accurate.
>
>
> [W2] The lack of high-frequency details in output is a common artifact of using losses like MSE as the only supervision. The artifact itself is not necessarily related to LoRA. We chose to keep the loss as simple as possible and did not explore complicated losses because achieving SOTA on intrinsics accuracy is not the purpose of this paper.
>
> - [W2.1] StyleGAN is a feed-forward model, unlike diffusion models. Because of the nature of GANs, StyleGANs only require single-step inference and hence there is no equivalent multi-step inference for StyleGAN.
>
> - [W2.2] We want to apply LoRA to the most information-intensive layers. Previous works (Hertz et al. 2023, Chefer at el. 2023 and Hong et al. 2023) have shown that diffusion model’s attention layers capture meaningful semantics, aggregate geometrically-grounded information and “reflect the overall composition” of the input (Hertz et al. 2023).  Thanks for raising this point. We have already updated the PDF to include the description of the motivation at Line 259-260.
>
>     **Reference:**
>     - Chefer, Hila, et al. "Attend-and-excite: Attention-based semantic guidance for text-to-image diffusion models." ACM Transactions on Graphics, 2023.
>     - Hong, Susung, et al. "Improving sample quality of diffusion models using self-attention guidance." Proceedings of the IEEE/CVF International Conference on Computer Vision. 2023.
>     - Hertz, Amir, et al. "Prompt-to-Prompt Image Editing with Cross-Attention Control." The Eleventh International Conference on Learning Representations, 2023.
>
>     [W2.3] This observation is indeed very interesting. The empirical evidence currently available is not enough for us to draw any conclusions. However, papers like *Localizing and Editing Knowledge in Text-to-Image Generative Models* find that knowledge “is not localized in isolated components”. Different attributes are scattered amongst different blocks in the UNet. This observation can be a potential explanation for the need for applying LoRA on all blocks.
>
>     **Reference:**
>
>     - Basu, Samyadeep, et al. "Localizing and editing knowledge in text-to-image generative models." The Twelfth International Conference on Learning Representations. 2023.
>
> [W3] The two arguments are not contradictory to each other. SG-XL achieves lower FID on the **FFHQ** dataset. SG-XL trained on ImageNet (which is a different model than the FFHQ one) has limited ability to generate realistic images, as shown in Fig. 6. ImageNet dataset is much larger and more diverse than FFHQ, therefore is a much harder task for SG-XL.
>
> [W4] The mitigation of the over-saturation issue in SD-2.1 multi-step is mainly due to the Zero SNR trick and the v-prediction objective, which is why we adopt these methods from Lin et al.(2023). The authors of *Beyond Surface Statistics: Scene Representations in a Latent Diffusion Model*, which we cite as Chen et al.(2023) in our paper, show they cannot extract any meaningful depth information from the encoder-decoder while it can be easily extracted from the UNet.
>
> **Reference:**
>     - Shanchuan Lin, Bingchen Liu, Jiashi Li, and Xiao Yang. Common diffusion noise schedules and sample steps are flawed. arXiv preprint arXiv:2305.08891, 2023.
>     - Yida Chen, Fernanda Viegas, and Martin Wattenberg. Beyond surface statistics: Scene representations in a latent diffusion model. arXiv preprint arXiv:2306.05720, 2023.

---

> ### Author Response · Authors · 2024-11-25
> **Looking Forward to Your Feedback**
>
> We sincerely appreciate the time and effort you put into reviewing our paper. Every question you raised has been carefully reviewed and addressed, with the changes reflected both in our detailed responses and the revised PDF. As the Reviewer-Author discussion phase ending soon, we are eager to hear any further comments you might have. If you have any additional questions, we would be more than happy to provide thorough explanations.

---

> ### Author Response · Authors · 2024-11-30
> **Follow-up of Responses to Your Review**
>
> Dear Reviewer w4yW,
>
> Following our previous response to your comments, we wanted to check if you have any additional questions or require any clarifications. We would greatly appreciate knowing if our responses have adequately addressed your concerns, or if there are any other aspects of the paper that need further explanation.Thank you for your time and valuable feedback.
>
> Sincerely,
>
> Authors of Paper 4684

---

> ### Author Response · Authors · 2024-12-03
> **Consideration for Score Adjustment Following Revisions**
>
> Thank you for your insightful comments and feedback. Your suggestions have helped us improve the paper. Every question you raised has been carefully reviewed and addressed, with the changes reflected both in our detailed responses and the revised PDF.
>
> If your view of the paper has evolved, we kindly ask that you consider updating your score.
>
> Thank you very much for your time.

---

### Official Review · Reviewer_1965 · 2024-11-03

**Soundness:** 2
**Presentation:** 3
**Contribution:** 2
**Rating:** 5
**Confidence:** 4

**Summary:**

This paper focuses on investigating the intrinsic knowledge encoded within various generative models, such as GANs, autoregressive models, and diffusion models. The study aims to uncover whether these models inherently capture fundamental scene properties, including Depth, Surface Normals, Albedo, and Shading. Through the use of Low-Rank Adaptation (LoRA), a lightweight technique that introduces minimal learnable parameters, the authors propose a model-agnostic approach to recover these intrinsic features.

**Strengths:**

1. Their findings reveal that a minimal amount of labeled data, sometimes as few as 250 images, suffices for the effective recovery of these intrinsic images.
2. The research claims a positive correlation between the generative quality of a model and the accuracy of its recovered intrinsic properties, which is interesting.

**Weaknesses:**

1. In Table 1, why do different generative models exhibit varying abilities to capture scene intrinsics without altering the generator head? For instance, even within the same model, such as SG-XL, all scene intrinsics (e.g., Normal, Depth, Albedo, and Shading) are recoverable for the FFHQ dataset, while none of these intrinsics can be captured for ImageNet. Are there any hypotheses regarding which properties of the models or datasets might influence their differing abilities to capture intrinsic features?

2. While the paper asserts that “enhancements in image generation quality correlate positively with intrinsic recovery capabilities”, this claim seems not convincing enough due to the following reasons:

a) Figure 2 attempts to validate the claim by showing a relationship between generated image quality (measured by FID) and recovery errors. However, it includes only three generative models (SG-XL, SGv2, and VQGAN), which represent GAN-based and autoregressive models but exclude diffusion models. This limited selection makes it challenging to empirically confirm the claim. It would be better to provide more generative models in Figure 2, including diffusion models as well. Moreover, discussing the technical reasons behind such correlations would strengthen the argument.

b) The claim lacks rigor due to inconsistencies. For instance, Figure 2 suggests that SG-XL outperforms SGv2 and VQVAE in generative quality, yet Table 2 shows SG-XL occasionally underperforming them, such as in Shading (on FFHQ). Moreover, factors beyond the generative model itself, like the dataset, also impact performance. For example, SGv2 performs worse in Depth but better in Shading when switching from FFHQ to LSUN Bedroom. Please provide a more in-depth statistical analysis of the correlation between generative quality and intrinsic recovery capabilities across all models and datasets.

3. One of the paper’s objectives is to demonstrate that “tiny new parameters and data are enough for intrinsic recovery”. However, as mentioned in this paper, several existing works (e.g., [1]) have already shown that parameter-efficient adaptation methods, like linear probes, can effectively extract intrinsic features such as depth. This work merely replaces linear probes with LoRA and demonstrates its effectiveness, which feels somewhat incremental. Besides, it would be better to discuss more, e.g., it would be helpful to discuss further -- for example, whether there are particular scenarios where LoRA outperforms linear probes or if LoRA provides any advantages beyond performance.

[1] Yida Chen, Fernanda Viegas, and MartinWattenberg. Beyond surface statistics: Scene representations in a latent diffusion model. 2023.

4. The paper notes a lack of ground truth data for certain maps (like albedo), which raises the question of whether using a light physics simulator or tools like Unreal Engine could provide high-quality ground truth labels. This approach could also enable the creation of more complex scenes and lighting conditions for more rigorous model testing. I am curious about the feasibility of incorporating such simulators or tools into this work, and whether this approach could substantially enhance the reliability of the results.

**Questions:**

Please refer to Weaknesses.

---

> ### Author Response · Authors · 2024-11-22
> **Authors' Response to Reviewer 1965**
>
> 1. This is indeed a very interesting phenomenon. As we mentioned at Line 365-366 and illustrated in Fig.6, SG-XL trained on ImageNet has significantly lower visual quality than its counterpart trained on a much more constrained dataset, FFHQ. Obvious geometric unrealism and lack of details can be easily spotted in the uncurated samples provided in Fig. 6. We believe that overall bad image generation quality is related to the failed intrinsics recoveries.
>
> 2.
>     A.
>         Thank you for the suggestion. Below we have added the results of the diffusion model LDM trained on FFHQ (in bold) to the table. The new results also follow the correlation perfectly. As reported by the paper, its FID score is 4.98. We use LoRA rank 8 and 1e-3 learning rate, which are the same as the other models in the table.
>
>         We have also updated Fig.2 in the PDF file.
>
>     **Normal**
>     |            |FID|L1 Error| Median Error| Mean Error|
>     | -------- | ------- | ------- | ------- | ------- |
>     | SG-XL  |   2.19 | 12.63 | 18.07 | 15.28|
>     | SG-v2 |     3.62| 13.87 | 19.60 | 16.93|
>     | **LDM**   |   4.98| 14.86 | 20.19 | 18.04|
>     |VQGAN| 9.6 | 16.33 | 20.97 | 19.97|
>
>
>     **Depth**
>     |            |FID|% delta>1.25|RMSE|
>     | -------- | ------- | ------- | ------- |
>     | SG-XL  |   2.19 | 6.13|0.1337|
>     | SG-v2 |     3.62| 9.26|0.1530|
>     | **LDM**   |   4.98| 17.7|0.1738|
>     | VQGAN | 9.6 | 37.67|0.1819|
>
>     B.
>     Given the nature of intrinsics recovery experiments for generated images, our analysis relies on off-the-shelf intrinsics predictors. While Paradigms (Forsyth & Rock, 2021; Bhattad & Forsyth, 2022) is among the most effective predictors we found for albedo and shading, it still falls behind modern predictors for depth and normals. This limitation makes the shading metrics less reliable as definitive indicators of the correlation. It must also be noted that albedo-shading decompositions are an ill-posed problem and there are no clear indicators yet of what is an ideal intrinsic image decomposition (see a recent work by Zhang et al NeurIPS 2024; which discusses the ill-posedness of intrinsic image decomposition).
>
>      Moreover, shading is inherently one of the “easier” intrinsics to predict, leading all three models to converge quickly to similarly low errors. Consequently, the occasionally reversed ranking for shading does not undermine the broader correlation observed between generative quality and intrinsic recovery capabilities.
>
>     We would also like to highlight that comparing metrics between models trained on different datasets is not meaningful due to variations in data distribution and geometric properties. This is similar to the fact that FID scores from different datasets cannot be directly compared.
>
>     **Reference:**
>         - David Forsyth and Jason J Rock. Intrinsic image decomposition using paradigms. IEEE transactions on pattern analysis and machine intelligence, 2021.
>         - Anand Bhattad and David A Forsyth. Cut-and-paste object insertion by enabling deep image prior for reshading. In 2022 International Conference on 3D Vision (3DV). IEEE, 2022.
>         - Zhang X, Gao W, Jain S, Maire M, Forsyth D, Bhattad A. Latent Intrinsics Emerge from Training to Relight. arXiv preprint arXiv:2405.21074. 2024.
>
> 3.
>     [1] (which we cited as Chen et al.) learns a head to project intermediate features in the SD UNet to depth or saliency map. Due to this nature, the resolution of the intrinsics in [1] is just 32x32, 1/16 x 1/16 of the original resolution of the input. Our method, as we pointed out in Line142, recovers intrinsics using the **same** output head as the original image generation task, thus keeping the original input resolution. In addition, [1]’s method is restricted to the diffusion model, while our method is more generic to different architectures and types of intrinsics.
>
>     As for comparison to linear probing, Sec. 4.4 in our paper compares LoRA to linear probing (similar to [1]) and full fine-tuning. We found LoRA recovers better intrinsics than other approaches.
>
> 4.
>     Thank you for the great suggestion. We do believe having more accurate ground truth will enhance the quality of the results. Exploring the most efficient way of incorporating physics engines into the pipeline can be an interesting topic for the follow-up work. However, as mentioned in the general response, this paper is not about can we recover SOTA maps but rather an introspective analysis of the generative models’ learned representations. Our findings are that a wide range of generative models from GANs, Autoregressive to Diffusion Models internally encode intrinsic images. These are emergent phenomena that we believe will help us perhaps build better intrinsic image predictors which is beyond the scope of current work.

---

> > ### Comment · Reviewer_1965 · 2024-11-27
> > **Reply to Authors**
> >
> > Thank you for your response. However, I still have a few concerns:
> >
> > 1. Based on my previous questions and the authors’ response, I strongly suggest adding more figures to illustrate that "image generation quality correlates positively with intrinsic recovery capabilities". Currently, Figure 2 only covers the FFHQ dataset with Surface Normal and Depth. However, Table 2 includes additional datasets (ImageNet, LSUN Bed, Open) and tasks (e.g., Albedo and Shading), which may not exhibit a strong positive correlation, as I previously pointed out.
> >
> > 2. As LoRA is a widely used method with demonstrated effectiveness in many previous works, achieving better performance using LoRA for fine-tuning is not surprising. This result mainly reaffirms that LoRA is a promising fine-tuning method, which is why I view the contribution of this part as incremental.

---

> > > ### Author Response · Authors · 2024-11-27
> > >
> > > Thank you for your reply. Below is our response:
> > > 1. We note that generating such figures requires access to all types of generative models pre-trained on the same dataset at a specific resolution, with published FID scores for consistent comparison. To the best of our knowledge, FFHQ@256 is currently the only dataset for which pre-trained checkpoints and published FID scores are available for all generative models considered in this work.
> > >
> > >     However, Fig. 2 is not the sole evidence supporting this correlation. As mentioned in Lines 449–455 and shown in Fig. 10, we have also demonstrated that the correlation holds true within a single architecture (e.g., Stable Diffusion v1.1, v1.2, and v1.5) trained on open-domain data. These findings further reinforce our claim.
> > >
> > > 2. As mentioned in our general response, the main goal of our paper is to address the four key questions raised in the abstract (Line 40-45), rather than to propose a novel method. LoRA is simply a tool we used to probe and analyze the underlying mechanisms of generative models. While we respect your perspective, we respectfully disagree with the claim that our contribution is incremental.
> > >
> > >     To our knowledge, no prior work has extensively studied diverse types of generative models (GANs, autoregressive models, and diffusion models) to demonstrate their encoding of classical computer vision concepts such as intrinsic images. Moreover, we are not aware of any work that employs a model-agnostic approach—such as LoRA—to recover these concepts across all generative model architectures. Demonstrating that generative models inherently know intrinsic properties and that these can be effectively recovered is itself a significant contribution, offering insights into how such models generate strikingly realistic images.
> > >
> > >     Additionally, our discovery of a positive correlation between generative quality and intrinsic recovery capabilities suggests that high-quality image generation may require models to be grounded in some form of physical reality. This insight, supported by results on FFHQ and Stable Diffusion’s different checkpoints, lays a foundational step toward understanding the intrinsic capabilities of generative models.
> > >
> > >
> > > Thank you again for your constructive feedback and for helping us improve our work.

---

> ### Author Response · Authors · 2024-11-25
> **Looking Forward to Your Feedback**
>
> We sincerely appreciate the time and effort you put into reviewing our paper. Every question you raised has been carefully reviewed and addressed, with the changes reflected both in our detailed responses and the revised PDF. As the Reviewer-Author discussion phase ending soon, we are eager to hear any further comments you might have. If you have any additional questions, we would be more than happy to provide thorough explanations.

---

> ### Comment · Reviewer_1965 · 2024-11-29
> **Reply to Authors**
>
> Thank you for your response. I greatly appreciate the authors' efforts to delve into and analyze some intrinsic properties of generative models. However, as this is an analytical paper, I will review it from a stricter perspective. A paper of this type needs to have conclusions that are robust, insightful, and thoroughly substantiated.
>
> Concerning the first point, Figure 2 indeed conducts experiments only on the FFHQ dataset, which, being limited to facial images, lacks broad representativeness. Moreover, while Figure 10 tests SD models trained on an open set, it only presents results for the Surface Normal task. Other tasks—Depth, Albedo, and Shading—are not conducted. However, according to the results in Table 2, the claim of correlation does not necessarily hold for the tasks of Albedo and Shading. This is why I hope to see more experimental results.
>
> Regarding the second point, since this is an analytical paper, I expect an exploration into the underlying causes of the phenomena observed. For instance, the paper poses the question ‘How small can the required learnable parameters and labeled data be to successfully recover this knowledge?’, Figure 9 shows optimal results with a data size of 4000, with performance dropping when the dataset size increases or decreases. Yet, the paper does not address the reasons behind this trend. If this were a methodological paper, briefly mentioning data size as a hyperparameter might be acceptable. However, as an analysis-oriented paper, the paper should probe deeper into these reasons rather than merely mentioning them in passing.
>
> Therefore, I will maintain the original rating.

---

> > ### Author Response · Authors · 2024-12-02
> >
> > We would like to thank you again for your valuable comments and for taking the time to provide detailed feedback. Your insights will be instrumental in guiding us to improve our work.

---

### Author Response · Authors · 2024-11-22
**General Response by Authors**

###### *This post serves as a general response to address common questions. For more detailed discussions, please refer to our individual response to each Reviewer’s original post, where all questions and concerns are thoroughly addressed.*

We sincerely thank all reviewers for their time and thoughtful feedback. We are grateful that the Reviewers found our approach “generic” and “contributing to our understanding of these (generative) architectures”.

Below, we address the most common questions raised by the Reviewers:
1. **Correlation Between Generative Quality and Intrinsic Recovery Accuracy:**
Several Reviewers requested deeper discussion and additional results to reinforce the claim of a positive correlation between generative quality and intrinsic recovery accuracy. We have updated the PDF with additional results for the diffusion model (LDM). While acknowledging exceptions like shading, we emphasize that overall trends strongly indicate a connection.


2. **Generalization Across Architectures and Intrinsics:**
We updated the PDF to clarify the generality of optimal settings and explained in the individual response why certain experiments, such as multi-step inference, are limited to diffusion models only. We also provided additional explanations to highlight the advantages of LoRA over linear probing and full fine-tuning.


3. **Novelty and Scope of the Work:**
As discussed in our Introduction (Line 38-45), our paper takes an introspective and curiosity-driven approach to understand the capabilities of generative models. We explore whether they manipulate abstract representations of the world or representations grounded in physical reality. By investigating these questions, we aim to shed light on the intrinsic knowledge encoded within generative models and their ability to recover meaningful representations.



    In particular, we demonstrate that LoRA can recover intrinsic properties using the same output head as the original generation task—a novel capability not shown in prior works. Importantly, our approach is **architecture-agnostic**, meaning it can be applied across various generative model types, further broadening its applicability.

    It is important to note that our work neither aims nor claims to achieve SOTA performance on intrinsic prediction tasks. Instead, our primary focus is to address fundamental questions about generative models as raised in our abstract:

    1. What intrinsic knowledge do generative models encode?
    2. Can we establish a general framework to recover intrinsic representations from them?
    3. How small can the required parameters and labeled data be?
    4. Is there a direct link between generative quality and recovered intrinsic accuracy?

    By addressing these questions, our study provides valuable insights into the underlying mechanisms of generative models and opens up a promising direction for future research into their interpretability and utility. Finally, to our knowledge, our work is the first to address these questions comprehensively.

4. **Typos and Choice of Words:**
    We have updated the PDF to fix the typo pointed out by the Reviewers and replaced the word that may cause confusion.


We are grateful for the Reviewers’ insightful questions and suggestions, which have significantly improved the quality of our paper. Thank you again for your valuable feedback.

---

### Meta-Review · Area_Chair_cAs1 · 2024-12-21

**Metareview:**

The paper received mixed reviews post rebuttal. The paper is an analysis-oriented one, studying the potential of image generative models for intrinsic image recovery, with LoRA-based fine-tuning. The AC checks all the materials, and while appreciating the additional efforts including diffusion model based results and clarifications during the rebuttal period, concerns still remain for the current draft. Specifically:

- The current title, while being attractive, has the risk of over-claiming contributions. Readers would expect a comprehensive (or at least solid) understanding about "what do they know", and "do they know things", across different generative models, different model architectures, different data, different probing strategies, different types of things (and not just on image intrinsics). While the current finding is valuable, the limited scope is not broad enough to support such a title. This is not directly mentioned by the reviewers.
- Other aspects of the analysis is also less solid. E.g., as reviewer 1965 points out: "Figure 9 shows optimal results with a data size of 4000, with performance dropping when the dataset size increases or decreases. Yet, the paper does not address the reasons behind this trend. If this were a methodological paper, briefly mentioning data size as a hyper-parameter might be acceptable. However, as an analysis-oriented paper, the paper should probe deeper into these reasons rather than merely mentioning them in passing." Reviewer w4yW and the AC both concur on this point.

Therefore, the AC does not recommend acceptance given the current status of the paper. Besides incorporating reviewer feedbacks, to further improve the paper for a clear, targeted study that presents good scientific value to the community, it is highly recommended to revisit the writing and experimentation part to make sure *all the claims are solidly backed up by either experimentation or external references*. Reviewer w4yW also suggests additional results to include -- it's not necessary to incorporate all of them, the authors should see what's feasible -- but the AC values paper with all claims substantiated and over-claims or over-sells minimized.

**Additional Comments On Reviewer Discussion:**

Please see the meta review for detailed reasoning after checking all the materials.

---

### Decision · Program_Chairs · 2025-01-22

Reject